# Tectonic processes, variations in sediment flux and eustatic sea level recorded by the 20 Ma-old Burdigalian transgression in the Swiss Molasse basin

Philippos Garefalakis[1], Fritz Schlunegger[1]

[1]Institute of Geological Sciences, University of Bern, Bern, CH-3012, Switzerland

Correspondence to: Philippos Garefalakis (philippos.garefalakis@geo.unibe.ch)

**Abstract**

The stratigraphic architecture of the Swiss Molasse basin, situated on the northern side of the evolving Alps, reveals crucial information about the basin's geometry, its evolution and the processes leading to the deposition of the siliciclastic sediments. Nevertheless, the formation of the Upper Marine Molasse (OMM) and the controls on the related Burdigalian transgression have still been a matter of scientific debate. During the time period from c. 20 to 17 Ma, the Swiss Molasse basin was partly flooded by a shallow marine sea, striking SW – NE. Previous studies have proposed that the transgression

occurred in response to either a rise in global sea level, a reduction of sediment flux or an increase in tectonically controlled accommodation space. Here, we re-address this problem and extract stratigraphic signals from the Burdigalian Molasse deposits that can be related to changes in sediment supply rate, variations in the eustatic sea level and subduction tectonics. To achieve this goal, we conducted sedimentological and stratigraphic analyses of several sites across the entire Swiss Molasse basin.

Field investigations show that the transgression and the subsequent evolution of the Burdigalian seaway was characterized by (i) a deepening and widening of the basin, (ii) phases of erosion and non-deposition during USM, OMM and OSM times, and (iii) changes in along-strike drainage reversals. We use these changes in the stratigraphic record to disentangle between tectonic and surface controls on the facies evolution at various scales. As the most important mechanism, roll-back subduction of the European mantle lithosphere most likely caused a further downwarping of the foreland plate, which we use

to explain the deepening and widening of the Molasse basin particularly at distal sites. In addition, subduction tectonics also caused the uplift of the Aar-massif. This process was likely to have shifted the patterns of surface loads, thereby resulting in a buckling the foreland plate and influencing the water depths in the basin. We use this mechanism to explain the establishment of distinct depositional settings and particularly the formation of subtidal-shoals where a bulge in relation to this buckling is expected. The rise of the Aar-massif also resulted in a re-organization of the drainage network in the Alpine

hinterland, with the consequence that the sediment flux to the basin decreased. We consider that this reduction in sediment supply amplified the tectonically controlled widening and deepening of the Molasse basin. Because the marine conditions were generally very shallow, subtle changes in eustatic sea level contributed to the formation of several hiatuses that

chronicle periods of erosion and non-sedimentation. These processes also amplified the tectonically induced increase in accommodation space during times of global sea level highstands. Whereas these mechanisms are capable of explaining the establishment of the Burdigalian seaway and the formation of distinct sedimentological niches in the Swiss Molasse basin, the drainage reversal during OMM-times possibly requires a change in tectonic processes at the slab scale, possibly including the entire Alpine range between the Eastern and Central Alps.

In conclusion, we consider roll-back tectonics as the main driving force controlling the transgression of the OMM in Switzerland, with contributions by uplift of individual crustal blocks (here the Aar-massif). In addition, the reduction of sediment flux was likely to have been controlled by tectonic processes as well when basement blocks became uplifted, thereby modifying the catchment geometries. Eustatic changes in sea level explain the various hiatuses and amplified the deepening of the basin during eustatic highstand conditions.

# 1 Introduction

Foreland basins and their deposits have often been used for exploring the tectonic evolution of their hinterlands, mainly because these basins are mechanically coupled with the adjacent mountain belts (Beaumont, 1981; Jordan, 1981; DeCelles, 2004). The formation of these foreland basins occurs through the flexural downwarping of the underlying lithosphere in response to loading, which results in the formation of a wedge-shaped trough where sediment accumulates (DeCelles and Giles, 1996; Allen and Allen, 2005). The shape of the foreland trough depends on the mechanical properties of the foreland plate (Sinclair et al., 1991; Flemings and Jordan, 1990; Jordan and Flemings, 1991), on the load of the sedimentary fill itself (Flemings and Jordan, 1990; Jordan and Flemings, 1991), and predominantly on the tectonic and geodynamic processes leading to changes in plate loading (Beaumont, 1981; Jordan, 1981; Allen et al., 1991; Sinclair et al., 1991; DeCelles and Gilles, 1996). Additionally, a foreland basin can either be occupied by a peripheral sea when sediment flux is lower than the formation rate of accommodation space, or by a fluvial system if the opposite is the case (Sinclair and Allen, 1992). A shift from terrestrial to marine conditions, for instance, can occur through a reduction in sediment flux, an increase in tectonically controlled subsidence rate (Sinclair et al., 1991), or a rise in the eustatic sea level (Reichenbacher et al., 2013). This has particularly been inferred for the North Alpine Foreland Basin, or the Molasse basin, situated on the northern side of the European Alps (Fig. 1a), which experienced a change from terrestrial to marine conditions during Burdigalian times c. 20 Ma ago (e.g. Matter et al., 1980; Pfiffner, 1986; Schlunegger et al., 1997a; Kempf et al., 1999; Kuhlemann and Kempf, 2002; Ortner et al., 2011). In the Swiss part, this transgression resulted in the establishment of a shallow-marine seaway, linking the Paratethys in the NE with the Tethys in the SW (Allen et al., 1985), and it is recorded by the deposition of the Upper Marine Molasse group (OMM; Matter et al., 1980; Allen, P., 1984). Although the history and geometry of the Burdigalian seaway and the related sedimentary processes are well known through detailed sedimentological and chronological investigations (e.g. Lemcke et al., 1953; Allen, 1984; Allen et al., 1985; Homewood et al., 1986; Doppler, 1989; Keller, 1989; Jin et al., 1995; Salvermoser, 1999; Strunck and Matter, 2002; Kuhlemann and Kempf, 2002; Reichenbacher et al., 2013), the controls on this transgression have still been a matter of ongoing scientific debates. Previous authors (e.g. Allen et al., 1985; Homewood et al., 1986; Keller, 1989; Strunck and Matter, 2002; Reichenbacher et al., 2013) proposed a combination of a reduced sediment flux and a rise in global sea level as possible mechanisms. However, thermo-chronometric data from the core of the Alps (Lepontine dome, Fig. 1; Boston et al., 2017) and structural work in the external Aar-massif (Fig. 1; Herwegh et al., 2017) revealed that the Burdigalian was also the time of major tectonic events including thrusting in the Aar-massif and tectonic exhumation of the Lepontine dome (Schmid et al., 1996) leading to changes in surface loads in the Alps. Accordingly, it was proposed that tectonic processes could also have controlled the transgression of the peripheral sea through downwarping of the foreland plate (Sinclair et al., 1991). It is possible that the Swiss part of the Moalsse basin bears key information to differentiate between these underlying mechanisms (eustacy, sediment flux, tectonics) because it is situated to the North of the Aar-massif and the Lepontine dome (Fig. 1).

The aim of this paper is to disentangle between tectonic processes, a reduction of sediment flux and changes in the eustatic sea level as controls on the transgression of the OMM in the Swiss Molasse. We analysed OMM outcrops and sections along

the entire Swiss Molasse basin at both proximal and distal positions relative to the Alpine front (Fig. 1a). To this extent, we measured palaeo-flow directions, and we explored the OMM deposits for their sedimentary facies and related depositional settings. We also determined the thickness and grain size of sedimentary bedforms, and we applied hydrological concepts to calculate palaeo-water depths based on the measured parameters. However, a chronological framework is absolutely required for correlating sections across a basin where facies relationships are strongly heterochronous, as it is the case for the Swiss Molasse basin (Matter et al., 1980). Therefore, we also re-assessed the temporal framework of the analysed sections through a compilation of previously published magnetostratigraphic and biostratigraphic data, and we correlated the individual sections across the basin.

## 2 Geological setting

### 2.1 The architecture and evolution of the Alps

The doubly-vergent Alpine orogen (Fig. 1) is the consequence of the late Cretaceous to present continent-continent collision between the European and Adriatic plates (Schmid et al., 1996; Handy et al., 2010). It comprises of a double vergent nappe stack where the material on the northern side was derived from the European and Adriatic plates, while the rocks on the southern side is only of Adriatic provenance. In the centre, the Alps expose a crystalline core of European origin referred to as the Lepontine dome and the external massifs (Fig. 1, e.g. Aar-massif; Spicher, 1980). At deeper crustal levels, the Alpine orogen is underlain by a thick crustal root made up of a stack of lower crustal material derived from the European continental plate (Fry et al., 2010; Fig. 1b). Beneath the core of the orogen, the c. 160 km-long (Lippitsch et al., 2003) lithospheric mantle slab of the European continental plate bends and thus downwarps the foreland plate towards the SE (Fig. 1b). This bending was mainly driven by slab load due to the relatively large density of the subducted lithospheric mantle in comparison with the surrounding asthenosphere as seismo-tomography data reveals (Lippitsch et al., 2003). On the northern side of the Alps, the structurally highest unit is made up of Austroalpine nappes that structurally overlie the Penninicum, which in turn are underlain by the Helvetic thrust nappes (Fig. 1a). The front of the Helvetic and Penninicum units is referred to as the basal Alpine thrust (Fig. 1b). On the southern side, the Alps are made up of the Southalpine thrust sheets that consist of crystalline basement rocks and sedimentary units of African origin. This fold-and-thrust belt is bordered to the South by the Po basin (Fig. 1a). The northern side of the Alps are separated from the southern side by the north-dipping Periadriatic line that accommodated most of the shortening during the Oligocene and the early Miocene by backthrusting and right-lateral slip (Schmid et al., 1996).

Recently, slab load has been considered as the major driving force of the subduction history of the European plate and for the exhumation of crystalline rocks (Kissling and Schlunegger, 2018). This also concerns the exhumation of the Lepontine dome (Fig. 1), where normal faulting along the Simplon detachment fault resulted in rapid tectonic exhumation of the dome between late Oligocene and early Miocene times with a peak recorded by thermo-chronometric data at 20 Ma (Fig. 1b; Hurford, 1986; Mancktelow, 1985; Mancktelow and Grasemann, 1997; Schlunegger and Willett, 1999; Boston et al., 2017). This was also the time, when the Aar-massif, situated on the European continental plate (Fig. 1b), experienced a period of

rapid vertical extrusion (Herwegh et al., 2017). Herwegh et al. (2017) and Kissling and Schlunegger (2018) proposed a mechanism referred to as roll-back subduction to explain these observations. According to these authors, delamination of crustal material from the European mantle lithosphere along the Moho resulted in a stacking of buoyant lower crustal rocks beneath the Lepontine dome and the Aar-massif forming the crustal root (Fry et al., 2010; Fig. 1b). These processes are

considered to have the maintained isostatic equilibrium between the subducted lithospheric mantle and the crust and thus the elevated topography (Schlunegger and Kissling, 2015). Additionally, they most likely balanced, through the stacking of the crustal root (Fry et al., 2010, Fig. 1b), the rapid removal of upper crustal material in the Lepontine dome at c. 20 Ma (Schlunegger and Willet, 1999; Boston et al., 2017). Delamination of crustal material has also been invoked to explain the contemporaneous rapid exhumation and northward thrusting of the Aar-massif along steeply dipping thrusts (Herwegh et al.,

2017). These processes were contemporaneous with (i) the reorganization of the drainage network of the Central Alps (Kuhlemann et al., 2001a; Schlunegger et al., 1998), (ii) the decrease in sediment flux to the basin, as revealed by sediment budgets (Kuhlemann, 2000; Kuhlemann et al., 2001a, 2001b), and (iii) the Burdigalian transgression in the Swiss part of the Molasse basin. We will thus refer to these processes when discussing the controls on the Burdigalian transgression within a geodynamic framework.

## 2.2 The architecture and evolution of the Molasse basin

The Molasse basin is approximately 700 km long and striking ENE – WSW from France to Austria, where it broadens from <30 km to a maximum width of c. 150 km (Pfiffner, 1986; Fig. 1a). It is limited to the North by the Jura Mountains and the Black-Forest- and Bohemian-massifs, and to the South by the basal Alpine thrust (Homewood et al., 1986).

Reconstructions of the evolution of the Molasse basin (Fig. 2a) have been the focus of many research articles over the past years (e.g. Matter et al., 1980; Homewood and Allen, 1981; Allen, P., 1984; Keller, 1989; Schlunegger et al., 1996; Sinclair, 1997; Kempf et al., 1999; Kuhlemann and Kempf, 2002; Ortner et al., 2011; Reichenbacher et al., 2013). This has resulted in the general notion that the large-scale subsidence history of the Molasse basin was closely linked with the geodynamic evolution of the Alps (Sinclair et al., 1991; Kuhlemann and Kempf, 2002; Pfiffner et al., 2002; Ortner et al., 2011;

Schlunegger and Kissling, 2015). The development of this basin as a foreland through has been considered to commence with the closure of the Alpine Tethys in late Cretaceous times (Lihou and Allen, 1996; Schmid et al., 1996). This was the time when subduction of the European oceanic lithosphere with a large density beneath the Adriatic continental plate started. Large slab-load forces resulted in a downwarping of the European foreland plate and the formation of a deep marine trough (Schmid et al., 1996), where sedimentation occurred by turbidites (Sinclair, 1997) on submarine fans (Allen et al. 1991;

Sinclair, 1997; Lu et al., 2018; Reichenwallner, 2019). This first period of basin evolution has been referred to as the Flysch stage in the literature (Fig. 2b; Sinclair and Allen, 1992). The situation changed at 35 – 32 Ma when the buoyant continental lithosphere of the European plate started to enter the subduction channel. Strong tension forces started to operate at the stretched margin of the European continental crust particularly beneath the Central Alps (Schmid et al., 1996), with the result that the subducted oceanic lithosphere of the European plate broke off (Davies and von Blanckenburg, 1995). The

consequence was a rebound of the European plate, a rise of the Central Swiss Alps and an increase in sediment flux to the Swiss Molasse basin (Sinclair, 1997; Kuhlemann et al., 2001a, 2001b; Willett, 2010; Garefalakis and Schlunegger, 2018), which became overfilled at c. 30 Ma (Sinclar and Allen, 1992; Sinclair, 1997). The subsequent post 30 Ma-stage of basin evolution has been referred to as the Molasse stage (Sinclair and Allen, 1992).

Molasse sedimentation occurred from c. 30 Ma onward and was recorded by two large-scale, transgressive-regressive mega-sequences (Fig. 2b; e.g. Sinclair, 1997; Kempf et al., 1997; 1999; Kuhlemann and Kempf, 2002; Cederbom et al., 2004; 2011). These two mega-sequences consist of four lithostratigraphic groups. The first mega-sequence comprises the Lower Marine Molasse (UMM; Diem, 1986) and the Lower Freshwater Molasse (USM; Platt and Keller, 1992), and the second mega-sequence consists of the Upper Marine Molasse (OMM; Homewood et al., 1986) and the Upper Freshwater Molasse
(OSM; Matter et al., 1980). Sedimentation in the Molasse basin continued up to c. 10 – 5 Ma, when a phase of uplift during Pliocene times resulted in erosion and recycling of the previously deposited Molasse units (Mazurek et al., 2006; Cederbom et al., 2004; 2011). This erosion reached deeper stratigraphic levels in the western part of the Molasse basin than in the eastern segment (Baran et al., 2014) with the consequence that the OMM deposits are only fragmentarily preserved in the West (Fig. 2a).

Sediment dispersal has changed during the Molasse stage of basin evolution. Prior to 20 Ma, during USM times, measurements of sediment transport directions (Kempf, 1998; Kempf et al., 1999) and sediment provenance analysis (Füchtbauer; 1964) revealed that the sedimentary material was transported to the East by braided to meandering streams. At that time, a coastline was situated near Munich, separating a deep marine trough farther east from a terrestrial environment to the west of Munich (Kuhlemann and Kempf, 2002). During OMM times, heavy mineral assemblages reveal that the Swiss
Molasse basin operated as a closed sedimentary trough, where all supplied material was locally stored (Allen et al., 1985). During OSM times, from c. 16.5 to 5 Ma, heavy mineral data imply that material with sources in the Hercynian basement north of Munich or the Bohemian-massif was supplied to the Swiss Molasse ("Graupensandrinne"; Allen et al., 1985; Berger, 1996), suggesting that material transport occurred towards the West (Kuhlemann and Kempf, 2002). The details of the reversal of the drainage direction have not yet been elaborated, and related scenarios lack a database with palaeo-flow
information, particularly for the OMM. The establishment of such a database and particularly assignment of a more precise age on the drainage reversal will be part of the scope of this article.

## 2.3 The Upper Marine Molasse

*2.3.1 Lithostratigraphic and sedimentologic framework*

The Upper Marine Molasse (OMM) deposits, which are the focus of this study, mainly consist of a suite of shallow marine sandstones and mudstones that were deposited between c. 20 – 17 Ma (Fig. 2b) in a c. 70 – 80 km-wide seaway (Allen and Homewood, 1984; Allen et al., 1985; Keller, 1989; Strunck and Matter, 2002). Close to the Alpine thrust front, the OMM-successions are up to 900 m thick and thin to a few tens of meters towards the distal basin margin farther northwest.

Large streams with sources in the Central Alps supplied their material to the Molasse basin, thereby forming megafans and conglomerate deposits with diameters >10 km that interfingered with the sea (Schlunegger et al., 1996; Kempf et al., 1999). Consequently, the facies relationships were strongly heterochronous across the basin, and terrestrial deposits, preserved as conglomerates of the OSM according to Matter (1964), grade into marine sediments of the OMM (Keller, 1989) over a lateral distance of few tens of kilometers. This is also the case in the study area, where thick conglomerate packages situated at the Alpine thrust front c. 50 km to the NE of the Aar-massif (Napf conglomerates; Matter, 1964; Haldemann et al., 1980) separate the basin into southeastern and northwestern segments with different lithostratigraphic schemes (Fig. 3a). For simplicity purposes, these areas will be referred to as the eastern and western segments in relation to the Napf. East of the Napf conglomerates, in the following text denoted as the Napf-units (Matter, 1964), the OMM has been grouped in two transgressive-regressive packages referred to as the Lucerne- and the St. Gallen-Formations (Keller, 1989). Both units comprise a suite of sandstones with mudstone interbeds. They are separated from each other by a m-thick palaeo-sol (Schlunegger et al., 2016). We will refer to these units as the OMM-I (Lucerne-Fm) and the OMM-II (St. Gallen-Fm), respectively (Fig. 3a). Keller (1989) additionally categorized the OMM-I east of the Napf-units into a lower wave-dominated unit and an upper unit where tidal processes are recorded, which we refer to as the OMM-Ia and OMM-Ib, respectively.

West of the Napf, the OMM at the proximal basin border has also been categorized in two units (Fig. 3a) but with a different scheme and different names. These are the Sense-Formation at the base (suite of sandstones with mudstone interbeds) and the Kalchstätten-Formation at the top (alternation of sandstones and mudstones). Further up-section, these marine deposits (Strunck and Matter, 2002) grade into the fluvial conglomerates of the Guggershorn-Formation and thus into the OSM (Strunck and Matter, 2002) (Fig. 3a). Magnetostratigraphic dating showed that the accumulation of the Guggershorn-Formation occurred contemporaneously with marine (OMM) sedimentation at more distal sites. These conglomerates thus represent the deposits of a braided stream that shed its material to the OMM sea, similar to the Napf conglomerates.

In the central basin near Fribourg (Fig. 2a), also on the western side of the Napf, sedimentological investigations of sandwaves (Homewood and Allen, 1981; Homewood et al., 1986) disclosed the occurrence of tidal bundles and a bi-directional dispersal of sedimentary material. These deposits have been assigned to a subtidal environment (Homewood and Allen, 1981) where material with sources in the Central Alps were re-distributed in the basin by strong tidal currents that entered the basin from the Tethys in the South and the Paratethys in the Northeast (Allen et al., 1985; Kuhlemann and Kempf, 2002; Bieg et al., 2008). Lithostratigraphic correlations suggest that the deposits at Fribourg most likely correspond to the lower Sense-Fm (Python, 1996) and thus to the OMM-Ia (see discussion, and Fig. 3a).

In the basin axis, between the Lake Neuchâtel and the Wohlen areas (Fig. 2a), coarse-grained sandstones with large-scale cross-beds where individual grains are larger than 2 mm have been interpreted as subtidal sandwaves (Allen and Homewood, 1984; Allen et al., 1985). These deposits are either calcareous-sandstones with shelly-fragments, referred to as the "Muschelsandstein" (Allen and Homewood, 1984; Allen et al., 1985, Fig. 3a). Alternatively, they occur as coarse-grained cross-bedded sandstones with large lithoclasts, also called the "Grobsandstein" (Jost et al., 2016).

*2.3.2 Chronostratigraphic framework*

Ages for the OMM deposits have been established by multiple authors through palaeontological analyses of mammalian fragments and teeth (Keller, 1989; Schlunegger, 1996; Kempf et al., 1999) and $^{87}Sr/^{86}Sr$ chemo-stratigraphy (Keller, 1989). The latter yield a numerical age between c. 18.5 and 17 Ma, particularly for the OMM-II on the eastern side of the Napf.

Subsequent magneto-polarity chronologies (Schlunegger et al., 1996; Strunck and Matter, 2002) paired with further micro-mammalian discoveries (Kempf et al., 1997; Kempf, 1998; Kälin and Kempf, 2009; Jost et al., 2016) allowed an update of the chronological framework (Fig. 3b) of Keller (1989) through correlations with the Magneto-Polarity-Time-Scale (MPTS) of Cande and Kent (1992, 1995) and the most recent astronomically tuned Neogene time-scale (ATNTS, Lourens et al., 2004). This yielded in the notion that the transgression of the peripheral sea and the deposition of the OMM started at

c. 20 Ma and was synchronous, within uncertainties, across the entire Swiss Molasse basin (Strunck and Matter, 2002). However, a temporal correlation of sections across the Napf, i.e. between eastern and western Switzerland (Fig. 3a), and a harmonization of the stratigraphic schemes (Fig. 3b) has not been achieved yet. This will be accomplished in this paper, and it will build the temporal framework for the discussion of the development of the basin.

**3 Methods**

**3.1 Location of sections and available database**

Following the scope of the paper, we established the facies relationships and sediment transport patterns during the transgressive phase of the OMM and thus mainly focused on the OMM-I. We proceeded through sedimentological investigations of key-sites (Fig. 2a), which expose the related succession in proximal (Entlen section east of the Napf-units;

Sense section west of the Napf-units), central (St. Magdalena site, Gurten drill core) and in distal positions (Lake Neuchâtel and Wohlen areas). The lithofacies in the Entlen and Sense sections was investigated in the field at the scale of 1:50. At each outcrop along these sections, data was collected as notes in the field book and hand drawings on digital photos (available from the senior author upon request). The results are then presented as logs on Figure 4, and in Tables 1 and 2. The St. Magdalena site and the Lake Neuchâtel and Wohlen areas only display outcrops rather than sections. Therefore, the

sediments at these locations have been sketched in the field and on digital photos, thereby paying special attention on collecting information about the orientation and thickness of cross-beds. The sedimentary material of the c. 200 m-deep Gurten drill core is not available. However, the sediments were photographed at high resolution at the University of Bern in 1989 (see Fig. S3 in Supplement). We used these photos to extract information on the lithofacies association encountered in the drilling.

**3.2 Reconstruction of sedimentary architecture**

The lithofacies have been identified (e.g. Schaad et al., 1992; Keller, 1989) based on the assemblage of sedimentary characteristics including: grain size, thickness, lateral extent if applicable, sedimentary structures, basal contact, colour and fossil content (Tables 1 to 5). The lithofacies types correspond to individual bedforms (see Tables 1 to 5 for references),

which bear information on flow strengths, flow directions, sediment supply and water depths (e.g., Keller, 1989). The combination of these parameters, usually recorded by distinct assemblages of lithofacies types, can be used to identify distinct sedimentary settings. Related concepts of facies analysis have been documented for fluvial deposits (Miall 1978; 1985; 1996; Platt and Keller, 1992) but are less standardized for shallow marine deposits. Here, we followed Keller (1989) and Schaad et al. (1992), who developed a concept for shallow marine deposits where lithofacies types are grouped into facies assemblages in a hierarchic order, based on which distinct shallow marine settings can be interpreted. We followed these authors and assembled the various lithofacies types into 5 depositional settings, which are from land to sea: Terrestrial, backshore, foreshore, nearshore and offshore. We then mapped the depositional settings at the scale of 1:25'000 at various sites across the Swiss Molasse basin where suitable outcrops were present (see Fig. 2a for visited sites).

**3.3 Determination of sediment transport directions**

Sediment transport directions were determined from orientations of clast imbrications, gutter casts, and dip directions of cross-beds. In addition, the orientation of the coastline can be inferred from sediment transport within the surf-and-swash zone at the wet beach where rolling grains carve mm-thin rills in the beach deposits, which are oriented perpendicular to the coast. These rills are recorded by linear grooves, or parting lineations, on the surface of sandstones (Allen, J., 1982; Hammer, 1984). We thus measured the orientation of these features where visible. We also determined the strike direction of oscillation-ripple marks to infer the orientation of waves and thus of the coastline.

**3.4 Calculation of palaeo-water depths**

We analysed the sediments in the key sections according to their palaeo-water depths. For oscillation-ripple marks, the ripple metrics (spacing between ripple crests and ripple heights) together with the grain size can be used to infer water depths at the time the oscillation-ripples were formed (Diem, 1985; Allen et al., 1985; Supplement). We thus measured the ripple metrics with a meter stick together with grain sizes in the field and calculated the water depths following Allen (1997). Likewise, minimum water depths can be inferred form heights of cross-beds as examples from modern streams have shown (Bridge and Tye; 2000; Leclair and Bridge, 2001). Please refer to the Supplement for the deviation of the related equations. Published information from deep-drillings (Boswil 1; Hünenberg 1; Schlunegger et al., 1997a) and seismo-stratigraphic data (Line 8307; Schlunegger et al., 1997a, Line BEAGBE.N780025; Fig. S2 in Supplement) completed the available database.

**4 Results**

**4.1 Proximal basin border to the East of the Napf: Entlen section (site 13 on Fig. 2a)**

The sedimentary suite of the c. 370 m-thick OMM-Ia at Entlen (Fig. 4a and Fig. S4a in Supplement) records a large diversity of lithofacies types (Table 1). Parallel-laminated (Sp), fine- to medium-grained sandstone packages are cm- to dm-thick and normally graded. These deposits alternate with dm-thick low-angle cross-bedded units with tangential lower boundaries (Sc, Sct$_a$) and layers with a massive structure (Sm). Gravel and pebble layers (Sg) and shell fragments (Shf) are visible where

sandstone units have erosive bases. Current- (Scr) and oscillation-ripple marks (Sos), locally with branching crests (Sbr), as well as flame-fabrics or sand-volcanoes (Sv) are present only in some places. Fine-grained lithofacies include mm- to cm-thick parallel-laminated to massive mudstone layers (Mp, Mm). Siltstone climbing-ripples (Mcl) are subordinate in the OMM-Ia suite. Mudstone drapes (Md), a few mm thick, mostly occur on top of current ripple-marks (Scr). In places, root-

casts are associated with yellow- to ocherish-mottled colours.

The overlying c. 430 m-thick OMM-Ib (Fig. 4a and Fig. S4a in Supplement) comprises fine- to medium-grained sandstone packages with mudstone interbeds (Table 1). Low-angle trough- (Sct$_r$) or tabular (Sct$_a$) cross-bedded sandstone beds are several decimeters thick. The Sct$_r$-sandstones contain current-ripple marks (Scr) at their base, whereas laminae-sets of Sct$_a$-sandstones are interbedded with current-ripples (Scr) recording opposing sediment transport direction. At one site, dm-thick

sandstone beds display a planar base and a wavy top with a wavelength of several meters (Spw; Fig. 4a). Parallel-laminated (Sp) and massive-bedded (Sm) sandstone beds are dm-thick and mainly found at the top of the OMM-Ib unit. Mudstones mostly occur as mudstone drapes (Md) on top of current ripple-marks (Scr). Lenticular- and flaser-interbeds (Mle, Mfl) are dm-thick and characterized by current ripple-marks with truncated crests. The OMM-Ib then ends with a m-thick mudstone displaying yellow to reddish mottling, root casts and caliche nodules.

Estimates of palaeo-water depths (see Supplement) reveal that the OMM-Ia sedimentary rocks were deposited in shallow conditions <5 m deep (Fig. 4a and Table S2). At the base of the OMM-Ib palaeo-water depths were >15 m and thus deeper than compared to the OMM-Ia unit (see Supplement). The OMM-Ib then shallows towards the top.

Measurements of bedform-orientations of the OMM-Ia deposits reveal sediment transport directions between 315° NW and 60° NE, with a dominant NE-directed transport (Fig. 4a). During OMM-Ib times, transport directions were bi-directional,

and measurements reveal the full range between 260° W and 70° E (Fig. 4a). Dominant transport directions of the OMM-Ib sediments change towards the N and to the W up-section.

## 4.2 Proximal basin border in the center: Napf-units (site 12 on Fig. 2a)

The Napf-units, which are a terrestrial interval of the OMM and the OSM (Fig. 3; Schlunegger et al., 1996), are c. 1550 m

thick and include a succession of conglomerates, sandstones and mudstone interbeds (Matter, 1964), which we categorize into 5 lithofacies types (Table 2, Fig. S4d in Supplement). Individual conglomerate beds are up to 10 m thick and display stacks of 2 – 3 m-thick beds with massive (Gm) to cross-bedded (Gc) geometries. The sandstone beds occur as massive-bedded (Sm) and cross-bedded (Sc) units. Interbedded mudstones are horizontally bedded (Mp) and have a yellowish-reddish mottling, caliche nodules and root cast. Palaeo-flow measurements imply a change from a NE-directed transport

during USM-times to a NW-directed sediment transport between OMM-I- and OSM times.

## 4.3 Proximal basin border to the West of the Napf: Sense section (site 7 on Fig. 2a)

The OMM at Sense (Fig. 4b and Fig. S4b) starts with a c. 200 m-thick succession of predominantly sandstones with some mudstone interbeds. Medium- to coarse-grained sandstone beds, up to 2 – 3 m thick, are massive-bedded (Sm), parallel-

laminated (Sp) and trough cross-bedded (Sct$_r$) (Table 3). They also occur as m-scale tabular cross-beds (Sct$_a$) forming several m-thick sigmoidal fore-sets (Sc) with top- and bottom-sets and pebbly lags (Sg). These packages are well exposed along a nearby road-cut (Heitenried, Fig. 2a; 46°49'27" N / 7°18'42" E; Fig. S4c in Supplement). Some of these Sct$_r$-cross-beds contain current-ripple marks (Scr), which are draped with a muddy layer (Md). Ripple marks also build up tabular sandstone bodies. They are either asymmetric (Scr) or symmetric (Sos) and may display branching crests (Sbr). In places, the sandstone bodies are highly bioturbated (Sf). Mudstone interbeds are 10 – 20 cm thick, massive- (Mm) to parallel-laminated (Mp) and strongly bioturbated (Mf).

The first-occurrence of 5 – 10 m-thick sandstone beds at 200 m stratigraphic level (Fig. 4b) marks a distinct shift in the stratigraphic record where several m-thick cross-bedded sandstone beds dominate the sedimentary succession. At this level, (i) 5 – 10 m-thick normally graded sandstone beds overlie an erosive base and display epsilon cross-beds (Sce); (ii) cross-bedded sandstones (Sct$_a$) are several meters thick and tens of meters wide, and individual laminae-sets are superimposed by current ripples (Scr) with an opposite flow direction than the cross-beds themselves; nearly all laminae-sets of cross-beds (Sc) are superimposed by mudstone drapes (Md); and (iii) medium-grained sandstones display ridge-and-swale bedform geometries (Spw) with a small amplitude of a few decimeters and a large wavelength of several meters. Some of these Spw-facies are occasionally covered with oscillation-ripple marks (Sos). The Sense section ends with an alternation of dm-thick mudstone beds (Mm) and m-thick massive- to cross-bedded conglomerates (Gm, Gc). These conglomerates then evolve towards an amalgamation of several m-thick, massive- and cross-bedded packages, characterizing the uppermost c. 50 m suite of the Sense section (Fig. 4b).

Estimates of palaeo-water depths range between 5 – 10 m (Fig. 4b and Table S4 in Supplement) during deposition of the lowermost 200 meters. Conditions were deepest at the 200 m stratigraphic level, reaching water depths in the range of up to 30 m (see Supplement). Measurements of sediment transport directions cover the range between c. 0° N and 90° E (Fig. 4b) at the base of the Sense section, which then changed to an axial, bipolar SW – NE-directed transport and to a W-directed transport towards the end of the section (Fig. 4b).

**4.4 Central basin: St. Magdalena site and Gurten drill core (sites 4 and 9 on Fig. 2a)**

The sandstones within a cave-system near Fribourg (St. Magdalena; Fig. 2a and Fig. S4c in Supplement) are medium- to coarse-grained and display an amalgamation of up to 1 – 3 m-wide, cross-bedded troughs (Sct$_r$) with current-ripple marks (Scr) at their base. The cross-bedded troughs and the ripple marks are both covered by mudstone drapes (Md). The amplitude of the troughs is in the range of several decimeters, whereas the cross-sectional widths span several decimeters to meters. The sandstones also occur as massive-bedded units (Sm). They are occasionally interbedded with current-ripple marks (Scr) draped with mudstone layers (Md). Basal contacts are erosive. Measurements of morphometric properties (St. Magdalena; Fig. 2a) allow an estimation of water depth which is in the range of c. 3 and 5 m (Table S1 in Supplement). Sediment transport directions measured at the St. Magdalena site reveal a WSW – ESE-dominated sediment transport.

In the nearby c. 260 m-deep Gurten (Fig. 2a) drill core (Fig. S3 in Supplement), OMM-Ia deposits occur as cross-bedded sandstones (Sc) topped with mudstone drapes (Md). These lithofacies associations (Table 4) are most abundant within the drill core and make up c. 200 m of the log. However, because drill cores offer limited information about the dimensions of the encountered sediments, we were not able to determine if cross-beds can be assigned to tabular-beds ($Sct_a$) or to troughs ($Sct_r$).

## 4.5 Basin axis in the West and the East: Lake Neuchâtel and Wohlen areas (sites 2 and 16 on Fig. 2a)

Calcareous, shelly-sandstones (Scc) occur in the basin axis and are an assemblage of various lithofacies. This Scc-facies association is made up of 5 – 10 m-thick, coarse-grained sandstone beds with low-angle cross-beds (Sc) that contain coquinas and shell-fragments (Shf) and pebbles (Sg) in places. Interbedded fine-grained sandstones contain current-ripple marks (Scr) recording an opposite flow direction relative to the cross-beds (Sc).

In the West (sites at Lake Neuchâtel area; Fig. 2a and Fig. S4c in Supplement), mapping shows that Scc-"Muschelsandstein" deposits are c. 5 m thick and record NNE- to NE-directed sediment transport. Foreset thicknesses of these deposits thin to <1 m towards the front of the Napf-megafan, where herringbone cross-beds imply SW and NE-directed, bi-modal sediment transport. At the NE margin of the Napf, these Scc-"Muschelsandstein" deposits grade into Slc-"Grobsandstein" units, which show m-thick tabular cross-beds ($Sct_a$) or dm-thick trough cross-beds ($Sct_r$) where individual troughs have m-wide diameters. Measurements of palaeo-flow directions reveal a SW- and SE-directed transport. These deposits are either time-equivalent sediments of the Scc-"Muschelsandstein" and thus contemporaneous with the OMM-Ib succession, or they mark the base of the OMM-II succession (Jost et al., 2016). Farther east near the Wohlen area (Fig. 2a and Fig. S4d in Supplement), foresets of "Muschelsandstein" cross-beds are 6 to 8 m thick, and in some locations up to 10 m thick as reported by Allen et al. (1985). Sediment transport directions were oriented towards the SSW covering the range between 230° SSW and 250° WSW and striking parallel to the topographic axis. Estimates of water depths of the Scc-"Muschelsandstein" reveal palaeo-water depths (Table S1 in Supplement) between 60 and 100 m.

## 5 Sedimentological interpretation

### 5.1 Entlen section

The OMM-Ia sedimentary rocks of the Entlen section are assigned to a backshore to upper nearshore realm, within a wave-dominated environment (Fig. 4a, and please see Table 1 for references). Records of waves are inferred from: (i) tabular, parallel-laminated and normally graded (Sp) sandstones, which are interpreted to represent sediments of the surf-and-swash zone near the wet beach where sedimentation occurs in the upper flow regime; (ii) low-angle cross-beds (Sc) with pebbly lags and shell fragments (Shf), which could reflect sand-reefs (or shoals), rip channel fills or storm layers, and (iii) oscillation- (Sos) as well as branching-ripple marks (Sbr) pointing to wave-activity (Fig. 4a). Gravels and pebbly-lags (Sg) are either evidence for high-energy storm events or for river inflow from the backshore. Finer-grained lithofacies, which are

either indicative of rapid sedimentation (Sv, Mcl; Table 1) or incipient pedogenesis (Mp, Mm; Table 1), are consistent with a shallow marine, wave-dominated environment.

The basal part of the OMM-Ib suite is assigned to a foreshore to lower nearshore setting, shaped by the combined effect of wave- and tidal-activity (Fig. 4a). This is inferred by the observation that current ripple-marks (Scr), which are situated on top of lamina sets of tabular- (Sct$_a$) and trough-cross-beds (Sct$_r$), point towards an opposite flow direction than the cross-beds themselves (Fig. 4a). Mudstone drapes (Md) on top of ripple marks together with lenticular- and flaser-interbeds (Mle, Mfl), are supportive evidence for a tidal environment (references in Table 1). The occurrence of waves, however, is inferred from parallel-laminated sandstones (Sp) with parting lineations, and ridge-and-swale (Spw) structures at the base of the OMM-Ib-suite. At the top of the OMM-Ib, massive sandstones (Sm) and mudstones with mottled colours, root casts and caliche nodules mark the presence of a backshore, possibly terrestrial setting.

The change from a nearshore, wave-dominated environment (OMM-Ia) to an environment with tidal records (OMM-Ib) was additionally associated with a deepening from <5 m to >15 m, at the base of the OMM-Ib, followed by a regressive sequence. We thus consider the base of this unit as the maximum-flooding surface (MFS), separating the OMM-I into a transgressive OMM-Ia unit and a regressive OMM-Ib succession (Figs. 4a and 5a).

## 5.2 Napf-units

We interpret the association of massive (Gm) to cross-bedded (Gc) conglomerates, and massive- to cross-bedded sandstones (Sm, Sc) as deposits within a braided river system (Table 2, please see references there). In such an environment, conglomerates are common records of active channels. Massive- (Sm) to cross-bedded (Sc) sandstones alternating with mottled mudstones were most likely formed on the floodplains bordering the network of braided channels, when bursts resulted in the accumulation of crevasse-splay deposits (Platt and Keller, 1992). Mudstone interbeds (Mp) with evidence for palaeo-sol genesis formed when channel belts shifted away from the axis of the section (Platt and Keller, 1992). This facies-association was mapped over tens of kilometers, both across and along strike of the basin orientation. It is thus assigned to an alluvial megafan (Schlunegger and Kissling, 2015), which was deposited by braided streams.

## 5.3 Sense section

We assign the OMM of the Sense section to a tidal-dominated environment where deltaic estuaries dominated the sedimentary facies (Fig. 4b and please see Table 3 for references). This is inferred from: (i) sigmoidal cross-bedded sandstones (Sc) with distinct top-, fore- and bottom-sets and pebbly-lags (Sg) that are indicative of a delta, and (ii) trough cross-bedded (Sct$_r$) and massive bedded (Sm) sandstones that could represent mouth-bar deposits where estuaries (or tidal intlets) end. In such environments, current-ripple marks (Scr) with mudstone drapes (Md) at the base of tabular cross-beds (Sct$_a$) point to rhythmic changes of tidal-current slack water stages within a subtidal setting, whereas ripple-marks (Sos) with branching crests (Sbr) and parallel-laminated sandstones (Sp) were most likely formed under the influence of waves close to the beach. Massive-bedded (Mm), parallel-laminated (Mp) and strongly bioturbated (Mf) mudstone interbeds are assigned to

a tidal flat that established on the landside margin of the delta. Towards the top of the section, the facies successively evolves into a fan delta setting. This is inferred from the observation that the sedimentary suite thickens and coarsens upwards and ends with massive- to cross-bedded conglomerates (Gm, Gc), suggesting progradation of a delta (e.g., Schaad et al., 1992).

The first-occurrence of 5 – 10 m-thick sandstone beds at 200 m stratigraphic level (Fig. 4b) records a remarkable increase of the water depth, when 5 – 10 m-deep tidal channels (Sce-sandstone beds with epsilon cross-beds) grade into several m-thick subtidal sandwaves ($Sct_a$) and nearshore tempestites (m-scale Spw-sandstones with ridge-and-swale geometries, see Table 3 for lithofacies and references). These two latter lithofacies ($Sct_a$, Spw) are interpreted to record the deepest palaeo-water depth in the Sense section, when water depths were in the range of up to 30 m. We consider this stratigraphic level to record the maximum-flooding surface (MFS) within the Sense section (Fig. 4b), and we will use it for correlation purposes with the OMM succession at Entlen (see discussion).

## 5.4 St. Magdalena site and Gurten drill core

The several m-thick outcrops near Fribourg are interpreted as subtidal shoal deposits, which accumulated within a tidal-dominated environment (Table 4, please see references there). This is inferred from: (i) m-scale cross-bedded troughs ($Sct_r$) with current-ripple marks (Scr) at their base (see also Homewood and Allen, 1981; for a similar interpretation). Alternatively, sandstone troughs ($Sct_r$) could be assigned to a mouth-bar environment, where massive-bedded sandstones (Sm) would represent records of rapid sedimentation. In contrast, mudstone drapes (Md) on top of the ripple-marks (Scr) and cross-beds (Sc) are formed during low-energy tides, or possibly during slack stages. Similar deposits ($Sct_r$, or possibly $Sct_a$) within the Gurten drill core could also be interpreted as sediments of subtidal shoals, however, due to limited exposure, interpretations are non-conclusive. Shallow palaeo-water depths are also inferred from estimates of water depths ranging between 3 and 5 m.

## 5.5 Lake Neuchâtel and Wohlen areas

We interpret the Scc-"Muschelsandstein" (Table 5) sediments (containing coquinas and shell-fragments (Shf) and pebbles (Sg)) to have been deposited within the topographic axis of the Burdigalian seaway, (see also Allen et al., 1985, and Jost et al., 2016, for a similar interpretation). Meter-scale cross-beds ($Sct_a$), locally superimposed by current-ripple marks (Scr), have been interpreted to reveal deposition under strong tidal currents (Allen et al., 1985). These deposits are thus assigned to offshore, tidal-dominated sandwaves, where sediment transport was NNE- (Lake Neuchâtel area; Fig. 2a) or SSW-directed (Wohlen area, Fig. 2a). In places, pebbly-lags (Sg) are interpreted as flood-related splays of gravels into the offshore setting, derived from the neighbouring Napf-megafan. In contrast, the coarse-grained sandstones (Slc-"Grobsandstein" sediments; Table 5) reveal similarities to the subtidal-shoal deposits encountered at the St. Magdalena site where trough cross-bedded sandstones ($Sct_r$) and tabular cross-beds ($Sct_a$) dominate the facies assemblages. However, the deposits in the Wohlen area are coarser-grained, and cross-beds (Sc) have larger diameters, but similar thicknesses. We relate the coarse-grained nature

of these deposits to the proximity of the Napf-megafan in the SW. The cross-beds with larger wavelengths and similar amplitudes possibly imply stronger currents compared to the subtidal shoal-deposits near St. Magdalena.

## 6 Discussion

### 6.1 Re-appraisal of chronostratigraphic framework

#### 6.1.1 Entlen section

No magnetostratigraphic data is available for the Entlen section, but a temporal calibration of the deposits can be achieved through indirect lines of evidence. This particularly concerns the reconstruction of an age constraint for the basal transgression, which is accomplished using two lines of evidence: First, the transgression post-dates the deposition of the USM, which terminated at C6An1 at the Fischenbach-section (Fig. 3b; Schlunegger et al., 1996). Second, based on stratigraphic interpretations of palaeo-flow direction data, Strunck and Matter (2002) considered that the transgression of the OMM progressed from the East towards the West, where the first marine sediments have been dated with C6r in the Sense section (Figs. 3b, see next section). An E – W transgression of the OMM is also seen in seismic line BEAGBE.N780025 (Fig. S2 in Supplement), where OMM deposits onlap onto the USM in a westward direction. Accordingly, the onset of the OMM at Entlen on the eastern side of the Napf predates the transgression at Sense farther west. Based on these arguments, we set an age of c. 20 Ma for the base of the OMM-I in the eastern Swiss Molasse basin (Figs. 3b and 5), which is consistent with Kälin and Kempf (2009). For the top of the OMM-I, we determine an age using the magneto-stratigraphy of the Napf-section (Fig. 3b) c. 10 km to the West of Entlen. This section includes an alternation of 6 reversed- and 5 normal-polarized magnetozones (Schlunegger et al., 1996). The lowermost, very long normally polarized interval (N1, Fig. 3b) includes the mammalian fossil site Hasenbach 1 recording an MN3b-age (Schlunegger et al., 1996) or even a lower-MN3b-age, as a revision of the mammalian material has shown (Kälin and Kempf, 2009). This allows a correlation of the normally polarized interval N1 with chron 5En of the MPTS (Cande and Kent, 1992, 1995) or the ATNTS (Lourens et al., 2004), respectively (Fig. 3b). Since the third reversed magnetozone of the Napf-section is very short (R2), and since the ATNTS chron 5D spans several 100 kyr and is thus quite long, it is most likely that a hiatus encloses C5Dr2 to C5Dr1 (Fig. 3b). In addition, because (i) the change from MN3b to MN4a has been calibrated with C5Dr2 (Jost et al., 2016), and since (ii) the base of the overlying OMM-II (Figs. 3b and 5) has been dated with MN4 (Keller, 1989), we suggest that the inferred hiatus coincides with the boundary between the OMM-I and the OMM-II (Figs. 3b and 5). This age assignment is consistent with magneto-polarity stratigraphy in the Molasse basin c. 70 km farther to the East (Kempf and Matter, 1999). It is also consistent with micro-mammalian investigations in the distal Molasse basin c. 30 km farther north where Jost et al., (2016) found that deposits spanning MN3b and MN4a are missing. Based on these constraints, we suggest that the top of the OMM-I correlates with C5Dr of the MPTS or C5Dr2 of the ATNTS, respectively, followed by a c. 0.5 Ma-long hiatus (Figs. 3b and 5). According to this correlation, the sediments recording the maximum-flooding conditions in the Entlen section are c. 19 Ma old.

*6.1.2 Wohlen area*

Correlations of the OMM deposits from the Entlen section to the Wohlen area was accomplished by Schlunegger et al. (1997a) through a seismostratigraphic analysis of the seismic line 8307 (please see Fig. 2a for trace of line). The seismic data shows that the OMM-I deposits onlap onto USM strata and then overlap this unit (Fig. 5b). Schlunegger et al. (1997a) correlated the OMM-I sequence with their Unit B in the Entlen section, which corresponds to the top of the OMM-Ia in our stratigraphic scheme. In addition, our field investigations and micro-mammalian data by Jost et al. (2016) revealed that the "Muschelsandstein" follows on top of the OMM-Ia and most likely corresponds in age to the OMM-Ib. Based on these arguments, we constrain the deposition in the distal basin to the time interval between c. 19 and 18 Ma (Fig. 5b). However, based on seismostratigraphic investigations of line 8307, Schlunegger et al. (1997a) proposed that sedimentation was interrupted at c. 18 Ma by a c. 0.5 Ma-long or possibly longer hiatus. This time span has later been specified through new micro-mammalian discoveries by Jost et al. (2016), who noted that a record of MN4a is missing in the Wohlen area and that the base of the OMM-II hosts mammalian fragments that correspond to MN4b. The interpretation of an inferred unconformity is additionally supported through observations of vadose cements (Allen et al., 1985) within the "Muschelsandstein", and through evidence for a thick palaeo-sol separating OMM-I from OMM-II in the Entlen section (see sect. 4 and 5). We use the occurrence of vadose cements and the palaeo-sol to propose that the uppermost beds of the OMM-Ib (including the "Muschelsandstein") were exposed to erosion, or non-sedimentation, after deposition. Furthermore, because the "Muschelsandstein" unit records the deepest water depth during OMM times at distal sites, we tentatively suggest that deposition of these mega-sandwaves started at the same time when the deepest conditions (MFS) were recorded within the Entlen section (Fig. 5b).

*6.1.3 Sense section*

Magnetostratigraphic data for the Sense section was presented by Strunck and Matter (2002). These authors placed the USM/OMM-boundary at this site within C6r of the MPTS (Cande and Kent, 1992; 1995), or alternatively of the ATNTS (Fig. 3b; Lourens et al., 2004). The subsequent alternation of normal and reverse magnetozones was correlated by these authors with chrons 6r through 5Dn of Cande and Kent's MPTS (1992; 1995), the latter of which corresponds to C5Dn1 of the ATNTS (Lourens et al., 2004). Following Strunck and Matter (2002), a possible hiatus prior to c. 17.7 Ma is likely to be recorded also within the Sense section (Figs. 3b and 5a). This correlation implies that the lower Sense-Formation corresponds to the OMM-Ia, whereas the upper Sense-Fm and the Kalchsätten-Fm are time equivalent units of the OMM-Ib (Figs. 3a and 3b). The topmost 50 m of the Kalchstätten-Fm, where the base was characterized by the first appearance of conglomerates, follows upon this hiatus and corresponds to the OMM-II in our scheme (Figs. 3 and 5a). This further implies that a hiatus separates OMM-I from OMM-II across the entire basin (Fig. 5a). In addition, subsequent sedimentation (base of OMM-II) progressed from the West to the East (Fig. 5a). In the same sense, the sediment packages recording the maximum-flooding conditions (MFS) have most likely the same age across the entire basin (Fig. 5a).

### 6.1.4 Lake Neuchâtel area

No micro-mammalian sites have been reported for the OMM deposits in the distal western Molasse basin. Therefore, we cannot provide further constraints on the history of sedimentation. However, our field inspections in the area of Lake Neuchâtel (Fig. 2a) show a sedimentary succession similar to that in the east, where amalgamated sandstone beds are overlain by the "Muschelsandstein" unit. Our field inspections also show that these calcareous, shelly-sandstones thin from c.< 10 m in the Lake Neuchâtel area in the West to a few meters towards the distal margin of the Napf-megafan, consistent with the results by Allen et al. (1985). Because of the architectural similarity between the "Muschelsandstein" deposits in the East and the West, we tentatively consider that deposition of the "Muschelsandstein" occurred synchronously across the entire Swiss Molasse basin.

## 6.2 Evolution of the Molasse basin

The chronostratigraphic framework together with the sedimentological data and palaeo-flow directions are used to propose a scenario of how the basin evolved through time. During USM-times (Fig. 6a), prior to the Burdigalian transgression, the basin was occupied by alluvial megafans at the proximal basin border, which gave way to an axially-directed channel-belt system in the distal basin (Fig. 6a; Kuhlemann and Kempf, 2002). Analysis of heavy-mineral assemblages by Füchtbauer (1964) and measurements of palaeo-flow directions in our study area (sole casts and cross-beds, this paper) and in eastern Switzerland (Kempf et al., 1999) revealed a NE-directed material transport towards the Munich region (Fig. 6a), which is consistent with the results of previous syntheses (e.g., Pfiffner et al., 2002; Kuhlemann and Kempf, 2002). In this area, the Molasse streams ended in a peripheral sea where neritic- to open-marine conditions prevailed (Kuhlemann and Kempf, 2002). Within the basin, a possible divide for sediment transport was situated somewhere SW of Geneva. We infer such a separation of sediment dispersal based on published sediment transport directions. In particular, south of Geneva, tidal cross-beds imply a sediment dispersal towards the South and thus towards the Tethys (Allen et al., 1991; Allen and Baas, 1993). To the Northeast of Geneva, however, our own measurements and data from Allen et al. (1985) reveal a NE-directed sediment transport to the Paratethys.

The palaeogeographic reconstruction based on our and published data for the period between c. 20 – 19 Ma is shown in Fig. 6b. It illustrates that the central part of the Molasse basin changed to a shallow-marine sea, which was c. 40 km wide at that time. Our estimates of palaeo-bathymetic conditions and sedimentological data (Figs. 4a and 4b) reveal that the water depths corresponded to a subtidal and nearshore setting. Nearshore to possibly offshore conditions (30 – 50 m) are recorded by subtidal mega-sandwaves (Allen and Bass, 1993) south of Geneva and by the predominance of sandstone-mudstone-alternations within the Boswil and Hünenberg drill cores in the NE (Wohlen area, Fig. 2a; Schlunegger et al., 1997a). Subtidal shoals, in up to 5 m-deep water, occupied the western part of the central Swiss Molasse (Fig. 6b). This was already proposed by Homewood and Allen (1981), and it is here confirmed by our sedimentological data and estimates of palaeo-water depths (see Supplement). Measurements of sediment transport directions from the shoal deposits reveal bi-modal, SW-NE-directed transport, with a dominant NE-orientation. This is particularly the case at the proximal basin border near the

Sense section (Fig. 2a) where deltaic foresets accumulated within an estuarine-setting (Fig. 4b). Mapping of depositional settings allowed us to trace the shoal deposits towards the northern tip of the Napf-megafan, from where the shoals narrow from c. 20 km to c. 10 km over a 70 km-long distance along strike. It thus appears that the shoals were deflected towards the topographic axis through a dominant NE-directed material transport (Fig. 6b). This interpretation is additionally supported by measurements of the transport directions of the Napf-megafan (i.e. clast imbrications) and the coastal deposits at the Entlen section (i.e. parting-lineation, cross-beds) pointing a material transport towards the NE (Figs. 4a and 5a). At the distal margin of the basin, field inspections show that beach sandstones gave way to subtidal-shoal deposits up-section. It thus appears that the Molasse basin between the Lake Neuchâtel and Wohlen areas (Fig. 2a) was a region of sediment export to the NE and to the SW. Material transport was most likely accomplished through strong tidal currents that entered the Swiss Molasse as two major tidal waves from the Tethys in the South and the Parathethys in the Northeast (Bieg et al., 2008).

The situation during c. 19 – 18 Ma (Fig. 6c) started with the time when the maximum-flooding surface was formed in the depositional record (MFS; Figs. 5a and 5b). The sedimentological data reveals that this time was characterized by a widening of the basin to widths up to 80 km, and it was dominated by offshore conditions in the topographic axis with water depths >50 m, as the "Muschelsandstein" deposits imply (Fig. 6c). There, cross-bed orientations (our measurements and data by Allen et al., 1985) and heavy mineral assemblages (Allen et al., 1985) reveal that sediment transport in the eastern basin axis (Wohlen area) occurred towards the SW, whereas sediment dispersal in the western basin axis (Lake Neuchâtel area) was directed towards the NE. We use this information to propose that a sedimentary depocenter established at the northern tip of the Napf-megafan. In addition, this megafan is interpreted to have experienced a backstepping (see sect. 6.3.3 for explanation). We infer such a scenario from the first appearance of a bi-modal E-W-orientation of material transport in the Entlen section (Fig. 5a). Because an E-W-oriented sediment transport requires a free-passage for tidal currents along the southern basin margin, the sea-side margin of the Napf-megafan had to step back to allow such a passage to form (Figs. 6b and 6c).

The palaeo-geographic situation shown in Fig. 6d comprises the timespan between c. 18 and c. 14 Ma and displays the evolution from the OMM-II to the OSM. The OMM-II period followed a period of erosion and non-sedimentation across the entire Swiss Molasse basin, as our re-assessment of the chronological framework of the OMM reveals (Figs. 5a and 5b). In addition, measurements of sediment transport directions reveal a SW-oriented sediment transport at proximal positions (Fig. 5a), which is consistent with the results of previous syntheses (e.g., Pfiffner et al., 2002; Kuhlemann and Kempf, 2002). This also implies that a possible E-W divide for sediment transport shifted towards the region near Munich, or even farther east. Similar to Kuhlemann and Kempf (2002), we infer such a scenario from the supply of material with sources in the Hercynian basement north of Munich (Fig. 1a) or the Bohemian-massif ("Graupensandrinne", Fig. 6d; Allen et al., 1985; Berger, 1996; see also sect. 2), which implies a westward tilt of the basin axis. This period ended with the progradation of the alluvial megafans during the time of the OSM.

In conclusion, this study confirms the results and syntheses of previous authors on the general sedimentation and material transport pattern during the deposition of the OMM. Nevertheless, our refinement of the chronological framework in

combination with additional sediment transport data allow us to specify some further details on the development of the transgression of the OMM. These include: (i) The establishment of the Burdigalian seaway was accompanied by both a deepening and widening of the basin. These mechanisms occurred contemporaneously and were associated with a northward shift of the topographic axis to the distal basin margin, where offshore and thus deepest marine conditions established at

19 Ma; (ii) the reversal of the sediment transport direction from an originally NE-oriented sediment dispersal to a SW-oriented sediment transport started sometime after 20 Ma and was completed at 18 Ma at the latest; and (iii) a wave-dominated coastline (with some tidal records) established on the eastern side of the Napf-megafan (Fig. 4a), whereas a tidal-dominated estuarine environment characterized the proximal coastal margin on the western side of the Napf (Fig. 4b). These variations in sedimentation pattern appear to explain why the lithostratigraphic framework differs between both regions (see

Fig. 3a).

## 6.3 Mechanisms associated with the transgression of the OMM

### 6.3.1 Reversal of the drainage direction

We relate the reversal of the drainage direction between the OMM-I and the OMM-II to tectonic processes operating at

deeper crustal levels beneath the Alps. This interpretation is guided by Pfiffner et al. (2002), who related changes in sediment dispersal within the Swiss part of the basin to a possible tilt of the foreland plate, caused by the westward shift of the Ivrea body. This tectonic unit comprises of mantle rocks with a high density (Fig. 1a) and could thus have influenced the deflection of the foreland plate (Pfiffner et al., 2002). While this mechanism is a viable explanation for the westward-directed tilt of the basin axis, we argue that a complementary driving force beneath the Eastern Alps is required to explain

the drainage reversal across the entire basin at least between Germany and Switzerland. We thus present a hypothesis of a possible geodynamic scenario to explain the 18 Ma-old change in the drainage direction in the next section, but we also acknowledge that this interpretation is speculative at this stage and warrants further investigations. Such an exploration, however, requires that the geodynamic processes between 33 and 30 Ma are also considered. At that time, the subducted European oceanic lithosphere was considered to have broken off beneath the Central Alps (Davies and von Blanckenburg,

1995; Schmid et al., 1996). However, underneath the Eastern Alps the European oceanic lithosphere remained attached to the continental plate as palinspastic restorations revealed (Handy et al., 2015). The consequence was the rise of the Alpine topography and a large sediment flux to the Swiss Molasse basin (Sinclair, 1997; Kuhlemann et al., 2001a; 2001b; Willett, 2010; Garefalakis and Schlunegger, 2018), which became overfilled at c. 30 Ma (Sinclar and Allen, 1992; Sinclair, 1997; see also sect. 2.1). East of Munich, however, the basin still remained underfilled until c. 20 Ma as evidenced by deep marine

sedimentation, where debris flows and proximal turbidites accumulated within the basin axis (Fertig et al., 1991; Malzer et al., 1993). We use these observations to propose that vertically-directed slab-load forces were still downwarping the foreland plate beneath the Eastern Alps to allow such a deep trough to form. In contrast, slab break-off beneath the Central Alps most likely caused a rebound of the foreland plate in Switzerland (Schmid et al., 1996; Schlunegger and Castelltort, 2016). We

interpret that the consequence was a stronger downward deflection of the European foreland plate beneath the Eastern Alps compared to the Central Alps, which could explain the east-directed sediment transport prior to c. 20 Ma (Fig. 6a).

Between c. 20 – 17 Ma, i.e. during OMM-times, a remarkable change was recorded in the Molasse basin. The eastern Molasse basin experienced a change from deep to shallow marine conditions (Kuhlemann and Kempf, 2002), and the entire basin recorded a reversal of the drainage direction from the E to the W (see above). We relate these shifts to a change in the pattern of slab-load forces underneath the Eastern and Central Alps. Particularly, in the eastern Molasse basin, the change from deep to shallow marine conditions could reflect a response to slab unloading through delamination, or break-off, of the subducted European lithosphere underneath the Eastern Alps (Ustaszewski et al., 2008), while roll-back subduction of the European plate beneath the Central Alps of Switzerland continued, as Kissling and Schlunegger (2018) proposed. This could have resulted in a rebound of the European plate beneath the Eastern Alps, whereas plate downwarping continued beneath the Central Alps. We interpret that these along-strike differences in the plate deflection caused a westward tilt of the foreland plate, which in turn could have controlled the drainage reversal. The reasons for the along-strike differences in the subduction mechanisms are not clear at this stage and could either be related to (i) inheritance related to the Mesozoic phase of rifting (Schmid et al., 2004; Handy et al., 2010), and (ii) differences in the mechanical strengths and rheological conditions of the foreland plate between the Swiss and the German/Austrian Molasse basins (Tesauro et al., 2009; 2013).

*6.3.2 Widening of the basin*

The drainage reversal occurred simultaneously with the widening of the basin in central Switzerland, as our chronological refinement shows. Because both changes occurred at the same time, one could infer a causality of the underlying controls. We refrain from such a view at this stage since we lack a detailed 3D restoration of the tectonic and geodynamic situation of the Alps/Molasse basin system for that time. Nevertheless, we consider that a possible control on the widening of the basin can be identified based on a cross-sectional view across the Central Alps and the related geodynamic processes during the Burdigalian c. 20 Ma ago (Fig. 7). At that time, we propose that the velocity of roll-back subduction beneath the Central Alps of Switzerland was likely to have accelerated. We justify this interpretation through the observation that (i) tectonic exhumation of the Lepontine dome (Fig. 1), accomplished through slip along the Simplon detachment fault (Mancktelow and Grasemann, 1997), occurred at the highest rates at that time (Boston et al., 2017; Schlunegger and Willet, 1999), and that (ii) the rapid rise of the Aar-massif (Fig. 1) also commenced at 20 Ma (Herwegh et al., 2017). Following the concepts of Kissling and Schlunegger (2018), these processes require a mechanism where several tens of km-thick buoyant crustal material was delaminated from the subducting European continental plate and accumulated within the crustal root (Fig. 7) within a short time period. In agreement with Kissling and Schlunegger (2018), we interpret that a rapid phase of roll-back subduction would also shift the basin axis to more distal sites. We use these mechanisms to explain the cross-sectional widening of the Molasse basin at 19 Ma and the northward shift of the basin axis (Fig. 7), thereby giving way to the deposition of the offshore "Muschelsandstein" (Figs. 6b and 6c). This lithofacies association has been mapped along the distal basin border adjacent to the external massifs only.

*6.3.3 Uplift of the Aar massif and establishment of a wave-dominated coast in the East and tidal records in the West*

We interpret that the different coastal morphologies between the eastern and western sides of the Napf-megafan (see sect. 5) was rather controlled by tectonic processes than by contrasts between the tidal waves from the Paratethys and the Tethys. We tentatively exclude a surface control at this stage because the pattern of tidal sandwaves in the basin axis is not reflected by a corresponding coastal morphology at the proximal basin margin. In particular, the "Muschelsandstein" cross-beds are thicker in the Wohlen area (up to 10 m) than in the Neuchâtel region (<10 m; see sections 4 and 5 and Supplement), implying stronger shear velocities in the eastern basin axis compared to the West. At the proximal basin margin, however, the coastal deposits on the eastern side of the Napf predominantly record the activities of waves, while compelling evidence for tidal activities are more abundant to the West of the Napf-megafan. Instead, because the establishment of wave and tidal-dominated shorelines occurred contemporaneously with the rise of Aar-massif (Herwegh et al., 2017), we interpret that this tectonic event could have influenced the distribution of the depositional settings at the proximal basin border. In particular, as we explain in the following paragraphs, the rise of the Aar-massif is likely to have resulted in a shift of surface loads (Fig. 7) and in a buckling of the foreland plate, which possibly influenced the water depths and the distribution of facies.

Structural mapping in the Aar-massif (Wehrens, 2015; Wehrens et al., 2017) has revealed that crustal blocks were rising along steeply SE-dipping thrust faults (Figs. 1b). This process was related to roll-back subduction of the European mantle lithosphere and the related delamination of crustal material, which resulted in the rise of the Aar-massif (Herwegh et al., 2017; Fig. 1b). This mechanism also lifted the topography surrounding the massif to higher elevations (Fig. 7), thereby forming a positive anomaly in the topographic load in the region. Sinclair et al. (1991) and Sinclair (1996) explored a possible stratigraphic response to topographic loading, associated with uplift of the Aar-massif (Sinclair et al., 1991), through the application of a linear elastic plate model where thrusting and erosion are dynamically coupled. In their model, the distance between the location of thrusting (Aar-massif) and the site in the basin where a signal is expected depends primarily on the flexural rigidity (or alternatively the elastic thickness or the *Te*-value) of the crustal rocks underlying the foreland basin (Sinclair, 1996). The flexural rigidity of the rocks beneath the Swiss Molasse basin has been quantified with an elastic thickness of c. 10 km using stratigraphic constraints (Sinclair et al., 1991). This estimate is particularly based on thickness gradients of accumulated Molasse deposits across a section from the distal basin border to the Alpine thrust front. This pattern, however, could have been influenced by upper crustal in-homogeneities (Waschbush and Royden, 1992) such as e.g. pre-existing faults (Pfiffner, 1986). This could explain why estimates of *Te*-values that are based on stratigraphic data are lower (Sinclair et al., 1991; Schlunegger et al., 1997b) than estimates that are based on the curvature of the entire European foreland plate from the distal Molasse border to the core of the Alps and even deeper (Pfiffner et al., 2002; Schlunegger and Kissling, 2015), where *Te*-values up to 50 km have been proposed (see discussion in Pfiffner et al., 2002). However, because the flexural response of the Molasse basin to local topographic loads was likely to have been influenced by inherited faults in the basement (Pfiffner, 1986), lower *Te*-values appear more appropriate (Waschbusch and Royden, 1992). Accordingly, if we consider a local, and thus upper crustal response to loading, characterized by a *Te*-value of

c. 10 km (Sinclair et al., 1991; Schlunegger et al., 1997b), then shifts in surface loads through the km-thick stacking of additional material in the Aar-massif is likely to have resulted in the formation of several tens of meters of supplementary accommodation space at the proximal basin border, as the models of Sinclair et al. (1991) predict. As a result, depocenters in the Molasse basin are predicted to step back to proximal positions (Sinclair et al., 1991), which is consistent with our

interpretation of the flow directions and the inferred backstepping of the Napf-megafan (see sect. 6.1 and Fig. 6). In addition, according to Sinclair (1996), upward-directed bulging of a few tens of meters is expected at the distal (forebulge) and at the lateral margins of the load (lateral bulge). The spacing between an expected lateral bulge and the location of the surface forcing ranges between 50 – 100 km (using a *Te*-value of 10 km), which is consistent with the distance between the Aar-massif and the inferred subtidal-shoals in the western Swiss Molasse basin (near Fribourg; Figs. 2a and 6b). Accordingly, we

suggest that the establishment of subtidal shoals at the northern tip and on the western side of the Napf is the consequence of this bulging (Figs. 6b and 6c). Because the plate had an eastward tilt at that time, as inferred from palaeo-flow directions, such a flexural signal could possibly not be recorded on the eastern side of the Aar-massif where the marine conditions were too deep (Figs. 6b and 6c).

We additionally use these mechanisms to explain the development of different depositional settings at the proximal basin

border of the Swiss Molasse basin. East of the Napf-megafan, a relatively high subsidence (rate of c. 340 m/Ma for OMM-Ia and c. 430 m/Ma for OMM-Ib, based on data in Fig. 4a) most likely resulted in a steeper submarine gradient compared to the West, where the inferred bulging possibly lowered and subdued the submarine slopes (subsidence rate of c. 285 m/Ma for both OMM-Ia and OMM-Ib, based on data in Fig. 4b). This could explain why the evidence for wave action is predominantly recorded along the eastern proximal steeper basin margin. Indeed, investigations on modern coasts have

shown that steeper coasts tend to promote the formation of larger waves (Flemming, 2011). In contrast, in the western Swiss Molasse, estuaries and tidal-channels could develop as the wave energy decreased in the subdued coastal landscape. Note, we cannot fully exclude that uplift along basement faults beneath the Molasse basin in western Switzerland (Spicher, 1980) shifted the peripheral sea to shallow water depths during OMM-times. If such a mechanism did occur, then it could have amplified the effects related to flexural bulging.

*6.3.4 Controls influencing changes in sediment supply*

The time around 20 Ma was also characterized by a continuous reduction in sediment flux from originally 25'000 km$^3$/Ma prior to c. 20 Ma to c. 15'000 km$^3$/Ma thereafter (Kuhlemann, 2000; Kuhlemann et al., 2001a, 2001b), which could have contributed, together with the tectonic widening of the basin, to the transgression of the peripheral sea in Switzerland

(Fig. 8). The mechanisms leading to this reduction in surface mass flux are not fully understood (Kuhlemann et al., 2002), and multiple hypotheses have been proposed including: (i) shifts towards a dryer palaeo-climate paired with a widespread exhumation of crystalline rocks with low bedrock erodibilities (Schlunegger et al., 2001); (ii) tectonic exhumation of the Lepontine through slip along the Simplon detachment fault, which occurred in response to rapid roll-back subduction (see above and Kissling and Schlunegger, 2018). Tectonic exhumation was considered to have shifted the drainage divide farther

to the North, thereby substantially decreasing the source area of the Molasse basin (Kuhlemann et al., 2001a); and (iii) uplift of the Aar-massif, which was considered to have resulted in a reorganization of the Alpine streams and which was also associated with a reduction of the source area of the Molasse basin (Kühni and Pfiffner, 2001). Except for the palaeo-climate hypothesis, all other mechanisms are ultimately linked to the tectonic processes we have outlined in the sections above.

*6.3.5 Controls related to changes in eustatic sea level*

Whereas tectonic processes are recorded in the arrangement of depositional settings in the entire Swiss Molasse basin, signals related to the eustatic changes of sea level are possibly recorded by several hiatuses. This particularly concerns the times of non-sedimentation between OMM-I and OMM-II, and between the OMM- and the OSM-phase, which we have elaborated in section 5.1 (Figs. 5a and 8). In this context, $\delta^{18}O$-values measured on benthic foraminifera have been used as proxy for establishing patterns of sea level changes (Miller et al., 1998). In particular, a shift to more positive values of the stable oxygen isotope $\delta^{18}O$ implies a growth of polar ice sheets, where lighter oxygen isotopes ($\delta^{16}O$) are preferentially stored (Zachos, 2001). As a consequence, global sea level most likely decreased (amplitude of drop is not really known) during shifts towards heavier (and thus more positive) isotopic records in planktonic organisms (Miller et al., 1998). These patterns have been reconstructed by Miller et al. (1996; 1998) at a high resolution. Shifts towards larger $\delta^{18}O$-values generally coincide with times when hiatuses are recorded in the Molasse basin (Fig. 8; see also Pippèrr and Reichenbacher, 2017; Sant et al., 2017). We thus suggest that drops in global sea level of a few tens of meters initiated a phase of non-deposition in the Swiss part of the Molasse basin, at least between OMM-I and OMM-II at c. 18 Ma, and between the OMM and the OSM. In contrast, phases of deposition appear to have occurred during periods when the global sea level was high. This was most likely the case during the deposition of the OMM-II at c. 17 Ma when the isotope data imply that the sea reached a maximum eustatic level at the least during the Burdigalian (Fig. 8). We acknowledge that a rising sea level could also have contributed to the transgression of the OMM-I and the establishment of the maximum-flooding surface (MFS), but the amplitude of change is much less compared to the OMM-II. Instead, we consider that the reduction in sediment flux and the changes in tectonic processes were exerting a stronger control.

**7 Summary and Conclusion**

In summary, we suggest that the Burdigalian transgression was related to a combination of a deepening and widening of the basin and a reduction of sediment supply rates, which we ultimately relate to tectonic processes in the Alpine hinterland. In this context, we consider that roll-back subduction was most likely responsible for the widening of the basin in the foreland and for the shift of the basin axis to distal positions. In addition, roll-back subduction of the European mantle lithosphere and delamination of crustal material most likely resulted in the rapid exhumation of the Lepontine dome (Boston et al., 2017) and the associated rise of the Aar-massif (Herwegh et al., 2017). These processes are interpreted to have triggered the change in the configuration of the drainage network (Schlunegger et al., 2001; Kühni and Pfiffner, 2001), with the consequence that the sediment flux to the basin decreased. This reduction in sediment flux, together with the tectonic widening of the basin, was

thus likely to have shifted the basin to underfilled conditions, which could have allowed the transgression of the peripheral sea in Switzerland (Fig. 8). In addition, shifts in surface loads, caused by the rise of the Aar-massif, resulted in flexural adjustments in the Molasse basin through buckling of the foreland plate. We suggest that this influenced the water depths within the basin, which could explain the development of distinct depositional settings and the formation of subtidal-shoals where a lateral bulge is expected. Because of the formation of shallow marine conditions, subtle changes in eustatic sea level contributed to the occurrence of several hiatuses (Sant et al., 2017). Whereas these mechanisms are capable of explaining the establishment of the Burdigalian seaway and formation of distinct sedimentological niches in Switzerland, the drainage reversal during OMM-times possibly requires a change in the tectonic processes at a scale that includes the subduction history of the entire mountain range, at least between the Eastern and Central Alps. Current explanations are still speculative and await the results on ongoing research in the framework of the AlpArray initiative. At this stage, we conclude that the geodynamic processes in the Alps include subduction mechanisms, delamination of crustal material and the uplift of the Aar-massif, reorganization of the drainage network and lower sediment fluxes, which are reflected in the Swiss Molasse basin through the establishment of shallow marine conditions and a shift of the topographic axis towards more distal sites at 19 Ma. Accordingly, the Burdigalian transgression in Switzerland most likely had a tectonic driving force, but with amplifications through responses occurring on the surface of the Alps and the Molasse basin.

**Author contribution**

F.S. designed the study. P.G. carried out the experiments, collected and interpreted the data with support by F.S.. The figures and photos were done/taken by P.G. with support by F.S. F.S. and P.G. wrote the text.

**Competing interest**

The authors declare that they have no conflict of interest.

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

# Tables and Figures

**Table 1: Lithofacies encountered in the Entlen section**

| Facies assemblages | Structures & bedforms | Depositional setting & references |
|---|---|---|
| Mcl, Mm, Mp, Mfl, Mle | Climbing ripples (Mcl), mudstone drapes (Md) and flaser- (Mfl) and lenticular-bedding (Mle) within parallel-laminated (Mp) and massive-bedded mudstones (Mm) | Wave-dominated environment: Backshore setting, where sediments were deposited within a swampy area. Root-casts, reddish mottling and caliche nodules represent palaeo-sol formation. *Keller, 1989; Miall, 1996 ; Daidu et al., 2013* |
| Md, Mfl, Mle, | Mudstone drapes (Md), flaser- (Mfl) and lenticular-bedding (Mle) | Wave-dominated environment with strong tidal influence: Backshore to nearshore setting, where Mle mostly form in the supratidal (mudflat) and Mfl form in the intertidal (sandflat) or alternatively in the subtidal, if ripple crests are fully preserved. *Keller, 1989 ; Shanmugam, 2003; Daidu et al., 2013* |
| Sct$_r$, Sct$_a$, Scr | Trough- and tabular-cross-beds (Sct$_r$, Sct$_a$) superimposed by current-ripple marks (Scr) which record an opposite flow direction. | Wave-dominated environment with strong tidal influence: Nearshore setting, deposits of (subtidal) sanddunes and sandwaves. *Baas, 1978; Allen and Homewood, 1984; Jost et al., 2016* |
| Sg, Shf, Sm, Sp, Sv | Pebbly-lags (Sg), Shell-fragments (Shf) within massive- to parallel-laminated sandstones (Sm, Sp), occasionally with sand-volcanoes (Sv) | Wave-dominated environment: Foreshore to nearshore setting within the beach area, deposited in the surf-and-swash zone. *Allen et al., 1985; Dam and Andreasen, 1990; Keller, 1990; Miall, 1996; Jost et al., 2016* |
| Sbr, Scr, Sos, Sc, Sm, Sp | Ripple marks (Sbr, Scr, Sos) and cross-beds (Sc) within massive-bedded and parallel-laminated sandstones (Sm, Sp) | Wave-dominated environment: Nearshore to foreshore setting, where ripple marks form beneath waves, while Sp form in the surf-and-swash zone (beach area) *Baas, 1978; Reineck and Singh, 1980; Clifton and Dingler, 1984; Allen, J., 1984; Keller, 1989; 1990* |
| Spw, Sc, Sos, Sm | Sandstone beds with a planar base and a wavy top (Spw), internally cross-bedded (Sc), superimposed by oscillation-ripple marks (Sos) within massive-bedded sandstones (Sm) | Wave-dominated environment: Nearshore to offshore setting, high-energetic storm deposits (tempestites) *Allen, J., 1982; 1984; Clifton and Dingler, 1984; Miller and Komar, 1980a; 1980b; Diem, 1986; Rust and Gibling, 1990* |

**Table 2: Lithofacies encountered in the Napf-units**

| Facies assemblages | Structures & bedforms | Depositional setting & references |
| --- | --- | --- |
| Gc, Gm | Cross- (Gc) and massive-bedded (Gm) conglomerates | Fluvial-dominated environment: megafan deposits within a braided river system. Gm, Gc form in active channels.<br>*Platt and Keller, 1992; Schlunegger et al., 1997a* |
| Sc, Sm, Mp | Cross- (Sc) and massive-bedded (Sm) sandstones with parallel-laminated mudstones (Mp) | Fluvial-dominated environment: megafan deposits within a braided river system. Sc, Sm from crevasse-splay deposits. Mp facies (often yellowish-reddish mottled with caliche nodules and root casts) is evident for palaeo-sol genesis on a floodplain.<br>*Allen, J., 1982; 1984; Rust and Gibling, 1990; Dam and Andreasen, 1990; Keller, 1990* |

**Table 3: Lithofacies encountered in the Sense section**

| Facies assemblages | Structures & bedforms | Depositional setting & references |
| --- | --- | --- |
| Gm, Gc | Massive- to cross-bedded conglomerates (Gm, Gc) | Fluvial-dominated environment: Terrestrial setting, where coarse-grained rivers deposited material. *Platt and Keller, 1992; Schlunegger et al., 1997a* |
| Sc, Sg, Sct$_r$, Sm | Cross-bedded sandstones (Sc) with top-, fore- and bottom-sets with pebbly lags (Sg), associated with trough-cross- (Sct$_r$) and massive-bedded sandstones (Sm) | Fluvial-dominated environment with tidal influence: Foreshore setting, where deltas, or alternatively estuaries, enter the sea: Sc and Sg mark Gilbert-Delta-type deposits, while Sct$_r$ and Sm mark mouth bar deposits (or alternatively: sanddunes). *Allen, J., 1982; 1984; Allen and Homewood, 1984; Rust and Gibling, 1990; Dam and Andreasen, 1990* |
| Gm, Gc, Mm, Sm, Sg | Massive- to cross-bedded conglomerates (Gm, Gc), associated with massive-bedded sand- and mudstones (Sm, Mm). Occasionally, pebbles only occur as isolated layers within sandstones (Sg) | Tidal-dominated environment with fluvial influence (river inflow): Nearshore setting. Terrestrial derived material washed into subtidal setting by high energetic floods. *Dam and Andreasen, 1990; Platt and Keller, 1992; Miall, 1996; Schlunegger et al., 1997a* |
| Mm, Mp, Mf | Massive-bedded (Mm) and parallel-laminated (Mp) mudstones with bioturbation (Mf) | Tidal-dominated environment: Backshore setting, deposits of the supratidal (mudflat). *Dam and Andreasen, 1990; Keller, 1990; Miall, 1996* |
| Sm, Mm, Mf, Sf | Strongly bioturbated (Mf, Sf) massive-bedded mud- and sandstones (Mm, Sm) | Tidal-dominated environment: Backshore to foreshore setting, deposits of mud- (supratidal) and sandflats (intertidal) *Dam and Andreasen, 1990; Keller, 1990; Miall, 1996; Nichols, 1999* |
| Scr, Md, Sct$_a$, Sf | Current-ripples (Scr) and tabular cross-beds (Sct$_a$) with mudstone drapes (Md), occasionally with heavily bioturbated sandstones (Sf) | Tidal-dominated environment: Foreshore setting, deposits of the intertidal (sandflat), where bioturbation occurs (Sf). Mudstone drapes record slack-water phases. *Baas, 1978; Reineck and Singh, 1980; Allen and Homewood, 1984; Shanmugam, 2003; Nichols, 1999* |
| Sc, Sce | Cross-bedded (Sc) sandstones, occasionally forming epsilon cross-beds (Sce) | Tidal-dominated environment: Foreshore to nearshore setting, deposits of a (meandering) tidal channel. *Allen, J., 1982; 1984; Frieling et al., 2009* |
| Sos, Sbr, Sp | Oscillation- and branching-ripple marks that grade into parallel-laminated sandstones | Tidal-dominated environment with strong wave influence: Foreshore to nearshore setting, deposits of the beach area (surf-and-swash zone) and the wave-transformation area. *Reineck and Singh, 1980; Clifton and Dingler, 1984; Allen, J. 1984; Keller, 1990* |
| Sct$_a$, Scr, Md, Sct$_r$, Sm | Tabular- (Sct$_a$) and trough-cross-bedded (Sct$_r$) sandstones, superimposed with current-ripples (Scr) and mudstone drapes (Md), associated with massive-bedded | Tidal-dominated environment with fluvial influence (river inflow): Foreshore to nearshore environment. Estuaries (Sct$_a$, Scr, Md) entering the sea, building up mouth-bar deposits or alternatively subtidal sanddunes (Sct$_r$, Sm). |

| Facies assemblages | Structures & bedforms | Depositional setting & references |
|---|---|---|
| | sandstones (Sm) | Yalin, 1964; Baas, 1978; Allen and Homewood, 1984; Dam and Andreasen, 1990; Jost et al., 2016 |
| Sct$_r$, Scr, Md | Trough-cross-beds (Sct$_r$) superimposed by current-ripple marks (Scr) and mudstone drapes (Md) | Tidal-dominated environment: Nearshore setting, deposits of (subtidal) sanddunes and sandwaves. Baas, 1978; Allen and Homewood, 1984; Shanmugam, 2003; Jost et al., 2016. |
| Spw, Sos, Sg | Sandstone beds with a planar base and a wavy top (Spw), superimposed by oscillation-ripple marks (Sos) and embedded with pebbles (Sg) | Tidal-dominated environment with wave influence: Nearshore to offshore setting, high-energetic storm deposits (tempestites) Reineck and Singh, 1980; Miller and Komar, 1980a; 1980b; Clifton and Dingler, 1984; Diem 1986; Miall, 1996 |

**Table 4: Lithofacies encountered at the St. Magdalena site & Gurten drill core**

| Facies assemblages | Structures & bedforms | Depositional setting & references |
|---|---|---|
| Sct$_r$, Scr, Sc, Sm Md | Trough-cross-beds (Sct$_r$) and cross-bedded sandstones (Sc) superimposed by current-ripple marks (Scr) and mudstone drapes (Md), often associated with massive-bedded sandstones (Sm) | Tidal-dominated environment: Nearshore setting, deposits of (subtidal) sanddunes (or mouth-bar deposits) and megaripples, We infer these deposits as sediments of subtidal shoals. Baas, 1978; Allen, J., 1982; 1984; Allen and Homewood, 1984; Rust and Gibling, 1990; Dam and Andreasen, 1990; Shanmugam, 2003 |

**Table 5: Lithofacies encountered in the Lake Neuchâtel and Wohlen areas**

| Facies assemblages | Structures & bedforms | Depositional setting & references |
|---|---|---|
| Scc, Sc, Scr, Shf, Sg | Calcareous, shelly-sandstones (Scc; "Muschelsandstein") are made up of cross-bedded sandstones (Sc), contain coquinas and shell-fragments (Shf) and pebbles (Sg) in places. | Tidal-dominated environment: Offshore setting, mega-sandwaves deposited under strong tidal-currents. Pebbly lags (Sg) are interpreted as pebbles flushed into the sea by flood-events. Baas, 1978 ; Allen et al., 1985; Rust and Gibling, 1990 ; Miall, 1996 ; Jost et al., 2016 |
| Sct$_a$, Sct$_r$, Slc | Coarse-grained sandstones (Slc, "Grobsandstein") with trough (Sct$_r$) and tabular (Sct$_a$) cross-bedded geometries. | Tidal-dominated environment: Nearshore to offshore setting, sandwaves, or alternatively sanddunes, similar to the subtidal-shoal deposits (see St. Magdalena site), however larger in diameters, but similar thicknesses. Allen and Homewood, 1984; Jost et al., 2016 |

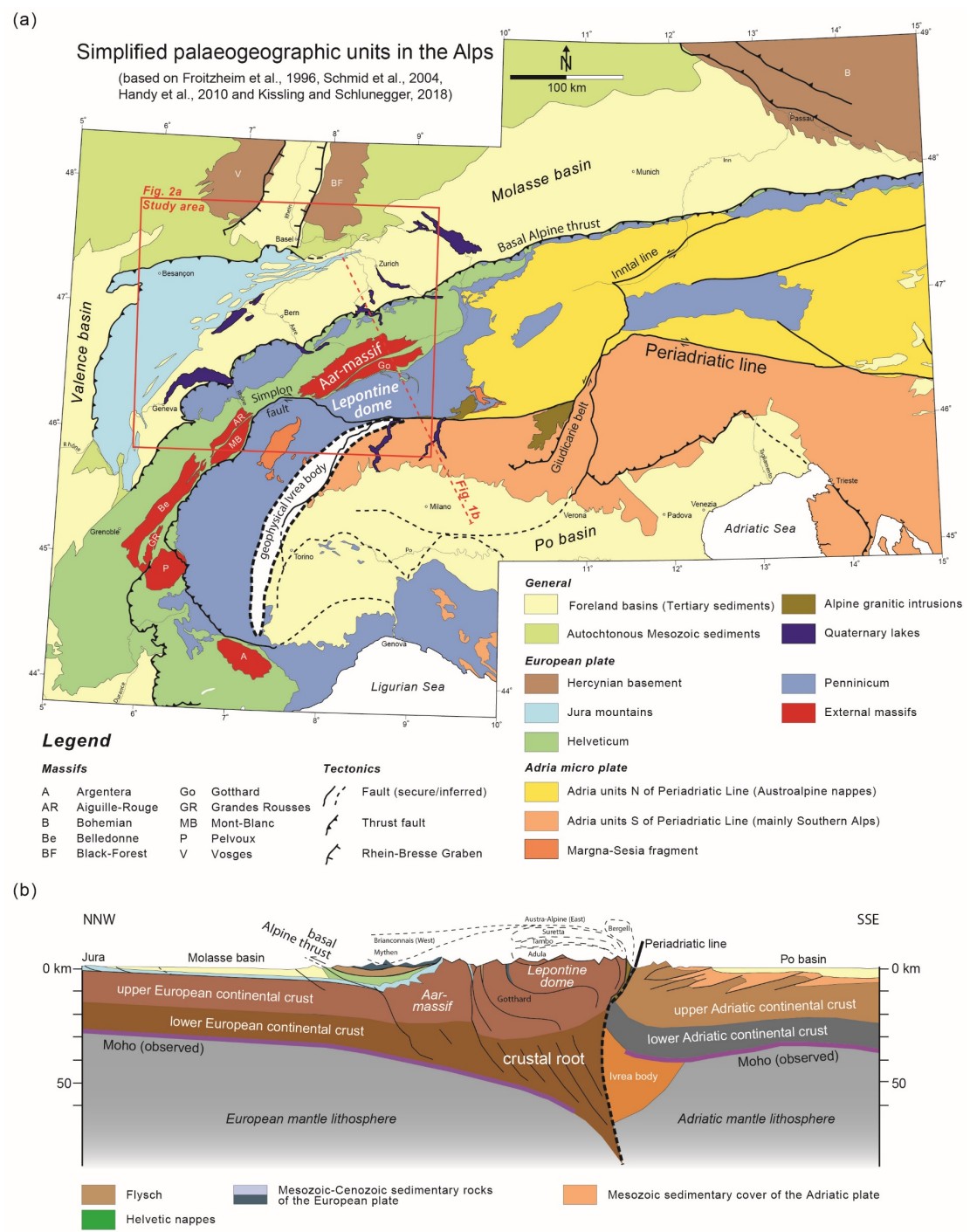

**Figure 1: a)** Simplified geological map of the European Alps based on a compilation by Kissling and Schlunegger (2018) and updated using additional information from Handy et al. (2015) and Pippèrr and Reichenbacher (2017); note location of Fig. 2a and trace of Fig. 1b. **b)** Simplified geological-geophysical section through the central European Alps adapted from Kissling and Schlunegger (2018).

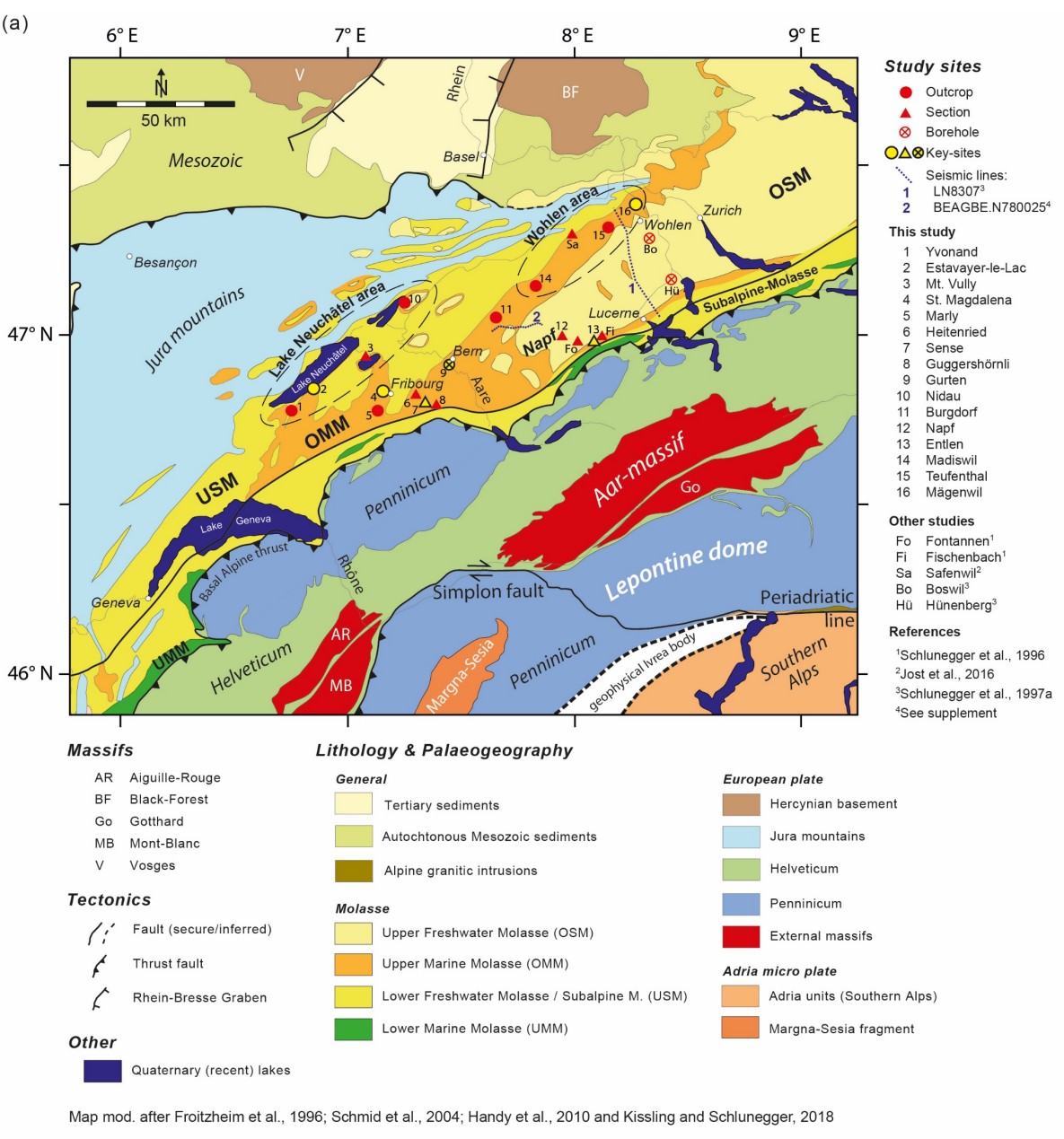

(a)

**Study sites**

- 🔴 Outcrop
- 🔺 Section
- ⊗ Borehole
- 🟡🔺⊗ Key-sites
- ⋯ Seismic lines:
  - **1** LN8307[3]
  - **2** BEAGBE.N780025[4]

**This study**

| 1 | Yvonand |
|---|---|
| 2 | Estavayer-le-Lac |
| 3 | Mt. Vully |
| 4 | St. Magdalena |
| 5 | Marly |
| 6 | Heitenried |
| 7 | Sense |
| 8 | Guggershörnli |
| 9 | Gurten |
| 10 | Nidau |
| 11 | Burgdorf |
| 12 | Napf |
| 13 | Entlen |
| 14 | Madiswil |
| 15 | Teufenthal |
| 16 | Mägenwil |

**Other studies**

| Fo | Fontannen[1] |
|---|---|
| Fi | Fischenbach[1] |
| Sa | Safenwil[2] |
| Bo | Boswil[3] |
| Hü | Hünenberg[3] |

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

**Massifs**

| AR | Aiguille-Rouge |
|---|---|
| BF | Black-Forest |
| Go | Gotthard |
| MB | Mont-Blanc |
| V | Vosges |

**Tectonics**

- Fault (secure/inferred)
- Thrust fault
- Rhein-Bresse Graben

**Other**

- Quaternary (recent) lakes

**Lithology & Palaeogeography**

*General*

- Tertiary sediments
- Autochtonous Mesozoic sediments
- Alpine granitic intrusions

*Molasse*

- Upper Freshwater Molasse (OSM)
- Upper Marine Molasse (OMM)
- Lower Freshwater Molasse / Subalpine M. (USM)
- Lower Marine Molasse (UMM)

*European plate*

- Hercynian basement
- Jura mountains
- Helveticum
- Penninicum
- External massifs

*Adria micro plate*

- Adria units (Southern Alps)
- Margna-Sesia fragment

Map mod. after Froitzheim et al., 1996; Schmid et al., 2004; Handy et al., 2010 and Kissling and Schlunegger, 2018

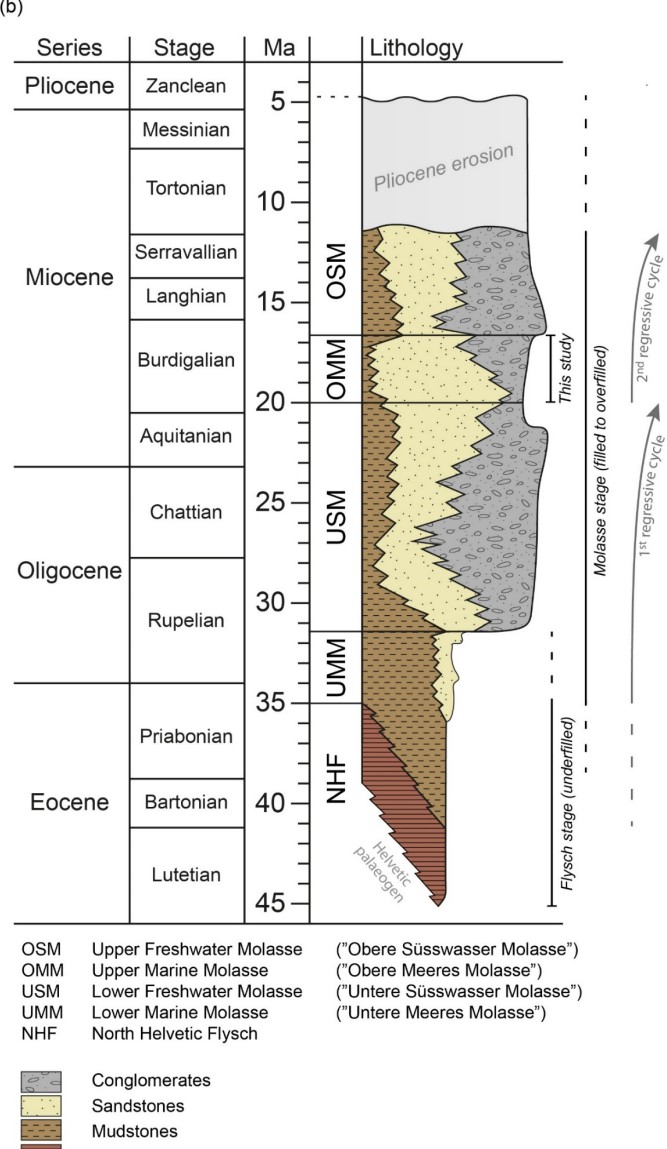

**Figure 2: a)** Detailed geological map of the area between Geneva and Zurich adapted from Kissling and Schlunegger (2018) showing the locations of data points referred to in this paper. The OMM deposits at sites 1 to 16 have been mapped at the scale of 1:25'000, which was used to reproduce Fig. 6. In addition, the observations of the sections, outcrops and the drill core at sites 2, 4, 7, 9, 12, 13 and 16 are explicitly described in sections 4 and 5 of this paper. Please refer to Fig. 1 for the complete legend. **b)** Lithostratigraphic scheme of the Molasse deposits in Switzerland. Modified after Keller (1989).

(a)

### Lithostratigraphic scheme to the West of the Napf

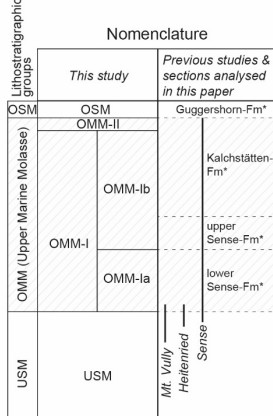

* Strunck and Matter, 2002

### Lithostratigraphic scheme at the Napf

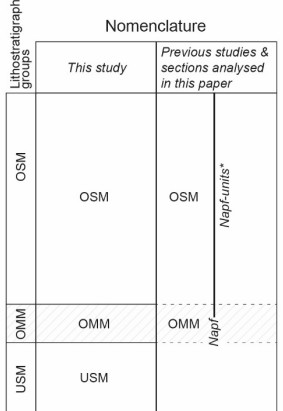

* Schlunegger et al., 1996

### Lithostratigraphic scheme to the East of the Napf

Lithostratigraphic groups

| | Nomenclature | |
|---|---|---|
| | This study | Previous studies & sections analysed in this paper |
| OSM | OSM | |
| OMM (Upper Marine Molasse) | OMM-II | St. Gallen-Fm* |
| | OMM-Ib | Lucerne-Fm* |
| | OMM-Ia | |
| USM | USM | Fischenbach / Entlen |

* Keller, 1989

### Lithostratigraphic scheme of sections in the distal basin

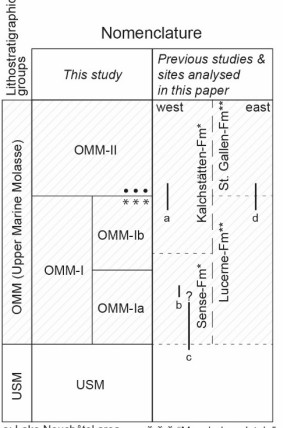

a: Lake Neuchâtel area
b: St. Magdalena site
c: Gurten drill core
d: Wohlen area

* Strunck and Matter, 2002   ** Keller, 1989

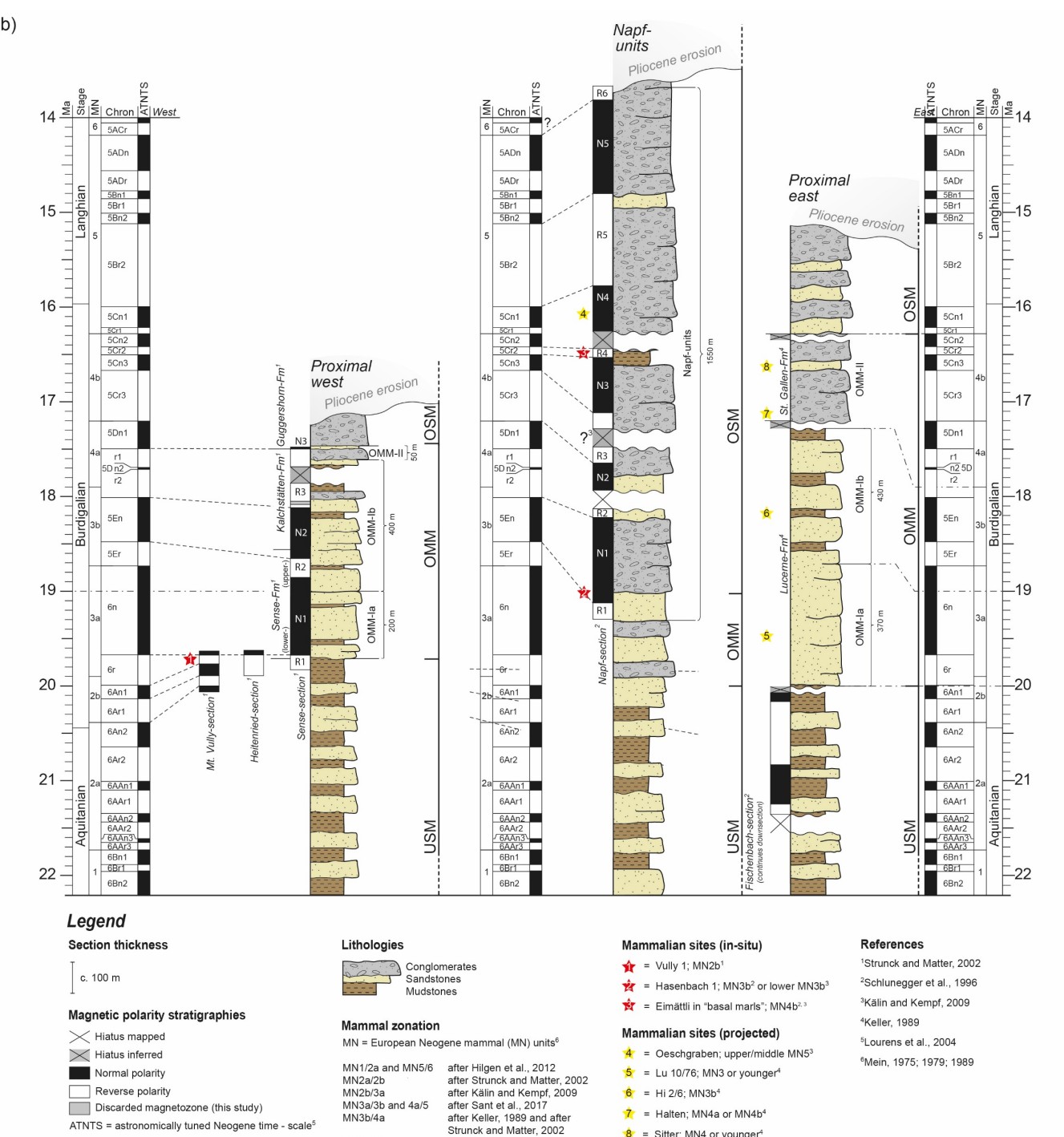

(b)

**Legend**

**Section thickness**

⌐ c. 100 m

**Magnetic polarity stratigraphies**

✕ Hiatus mapped
⊠ Hiatus inferred
■ Normal polarity
□ Reverse polarity
▨ Discarded magnetozone (this study)

ATNTS = astronomically tuned Neogene time - scale[5]

**Lithologies**

Conglomerates
Sandstones
Mudstones

**Mammal zonation**

MN = European Neogene mammal (MN) units[6]

| MN1/2a and MN5/6 | after Hilgen et al., 2012 |
| MN2a/2b | after Strunck and Matter, 2002 |
| MN2b/3a | after Kälin and Kempf, 2009 |
| MN3a/3b and 4a/5 | after Sant et al., 2017 |
| MN3b/4a | after Keller, 1989 and after Strunck and Matter, 2002 |
| MN4a/4b | after Jost et al., 2016 |

**Mammalian sites (in-situ)**

☆ = Vully 1; MN2b[1]
☆ = Hasenbach 1; MN3b[2] or lower MN3b[3]
☆ = Eimättli in "basal marls"; MN4b[2, 3]

**Mammalian sites (projected)**

4 = Oeschgraben; upper/middle MN5[3]
5 = Lu 10/76; MN3 or younger[4]
6 = Hi 2/6; MN3b[4]
7 = Halten; MN4a or MN4b[4]
8 = Sitter; MN4 or younger[4]

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

**Figure 3: a)** Lithostratigraphic scheme of the OMM in Switzerland. **b)** Composite stratigraphic columns illustrating sedimentary architectures at the proximal basin border in the western Molasse basin (proximal West), in the central part of the Molasse basin (Napf-units) and in the eastern basin (proximal East). The composite section for the proximal western basin is based on data from the Mt. Vully- and Heitenried-sections, drillings, and from surface information from the Sense section (Sense-beds and Kalchstätten-Formation) (Strunck and Matter, 2002). The composite section representative of the central part of the Molasse basin (Napf) is mainly based on the sedimentary logs by Schlunegger et al. (1996; see their Schwändigraben- and Fontannen-sections) complemented with information from the geological map of the region (Schlunegger et al., 2016). Note that Kälin and Kempf (2009) proposed a very short hiatus recorded by magnetozone R3 within the Napf-units, which we do not discuss in detail for simplicity purposes. The composite section illustrating the situation at the proximal basin border east of the Napf represents the sedimentary architecture as far east as of Lake Zurich (Fig. 2a). It is based on data from Keller (1989, see his Rümlig-, Ränggloch- and Lucerne-sections) and from Schlunegger et al., (1996, see their Fischenbach-section) and geological maps of the region (Wolhusen; Isler and Murer, 2019). Note that the Entlen section is situated immediately east of the Napf (Fig. 5a) where the lowermost part (Lucerne-Formation) can be characterized by the composite section of the proximal East. Detailed sedimentological data of the Sense-Formation and the Lucerne-Formation are shown in Fig. 4. Note that the Molasse units shown in capitals (i.e. USM, OMM and OSM) are based on the lithological architecture and thus on facies associations identified in the field.

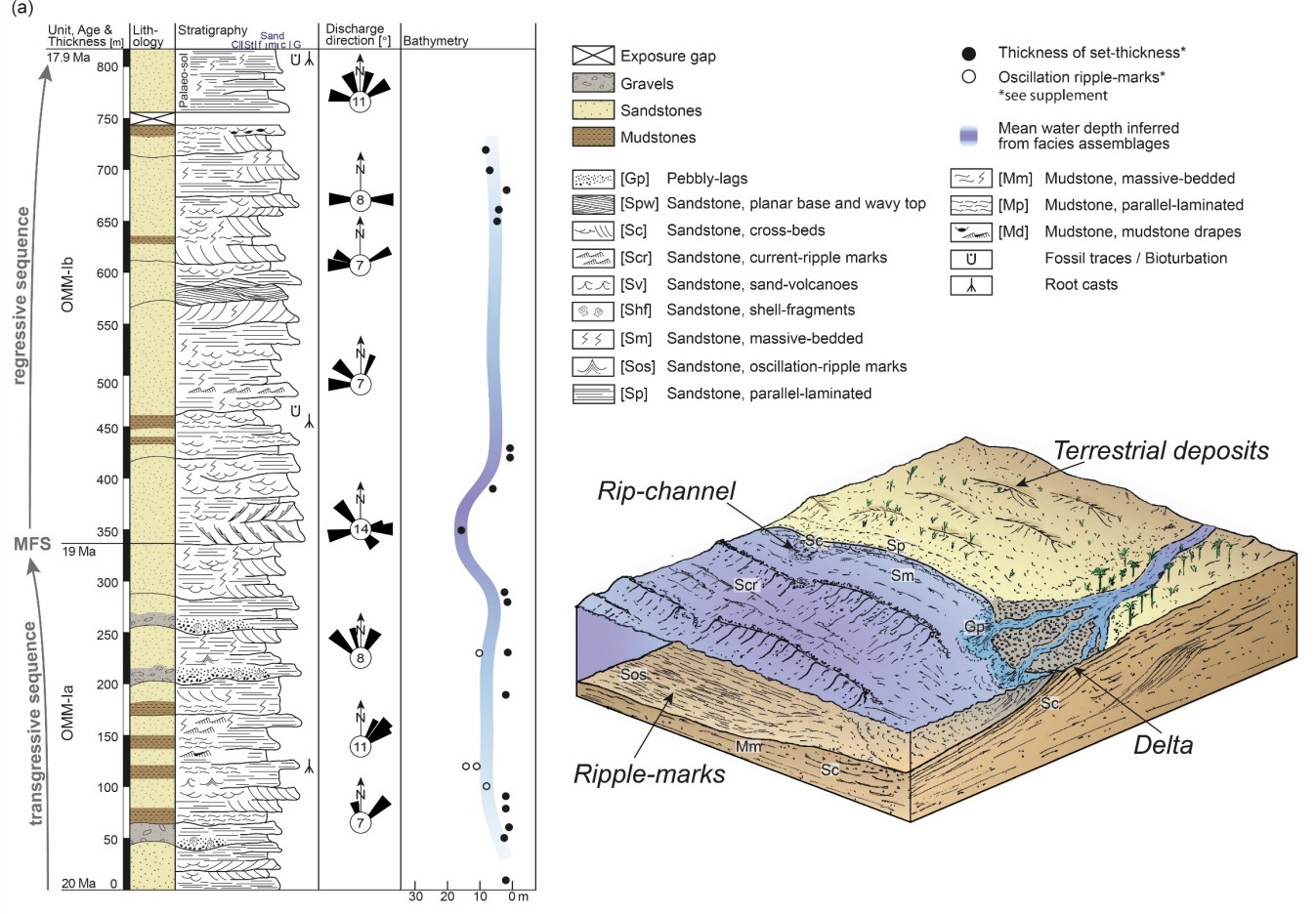

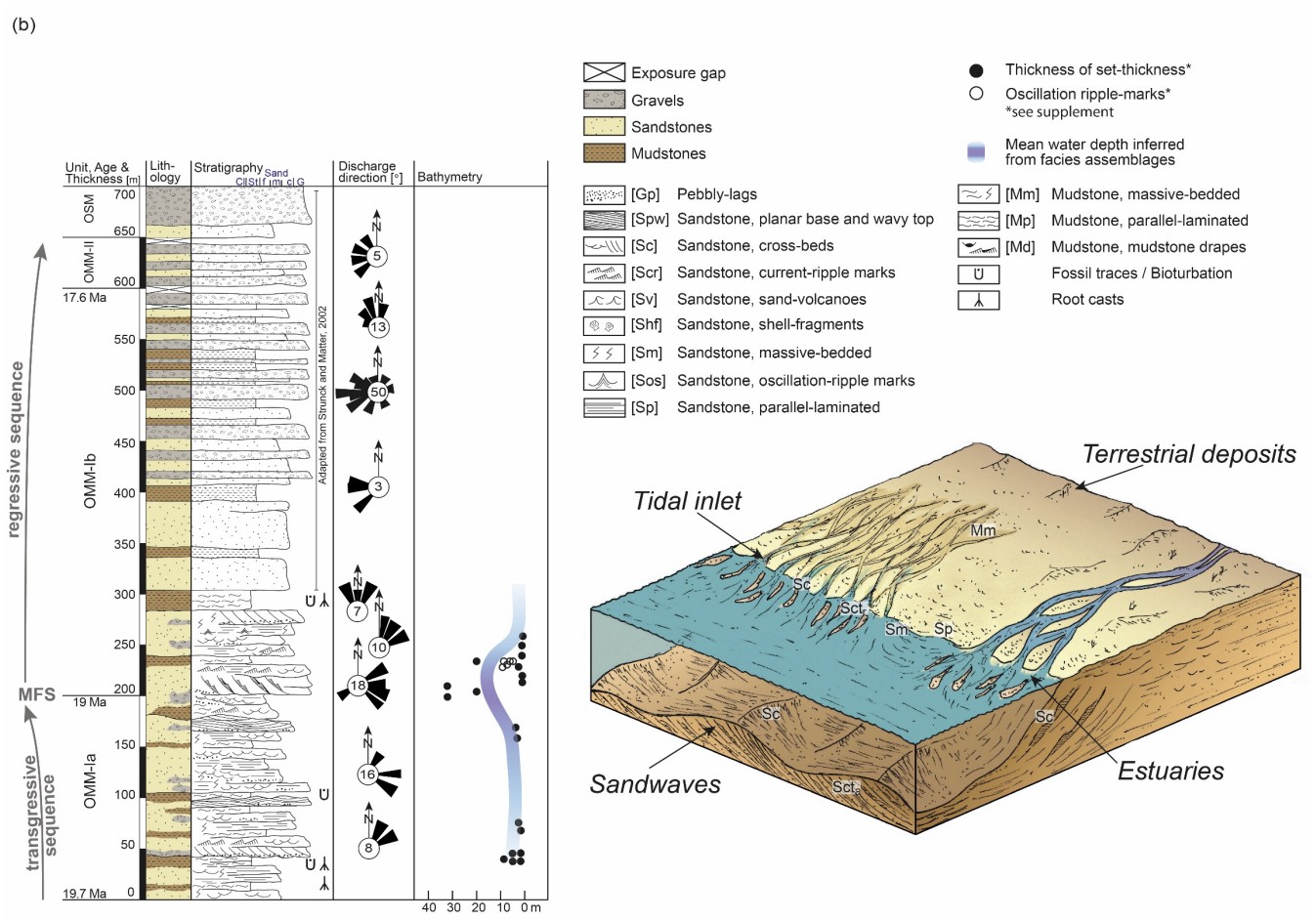

**Figure 4:** Sedimentological logs of **a)** the Entlen and **b)** the Sense section. See Fig. 2a for locations of sections, Fig. 5 for chronological framework of the deposits and the tables for further sedimentological details and abbreviations of the lithofacies, and for references to sedimentological work. The block-diagrams illustrate the palaeo-geographical conditions from a conceptual point of view. Note that the palaeo-bathymetric values are minimum estimates and that the mean water depths have been inferred from the assignment of lithofacies to a depositional setting. This might explain why the numerical values for water depths based on cross-bed thicknesses and our inferred mean water depth estimates deviate between c. 200 m and 250 m of the Sense-section.

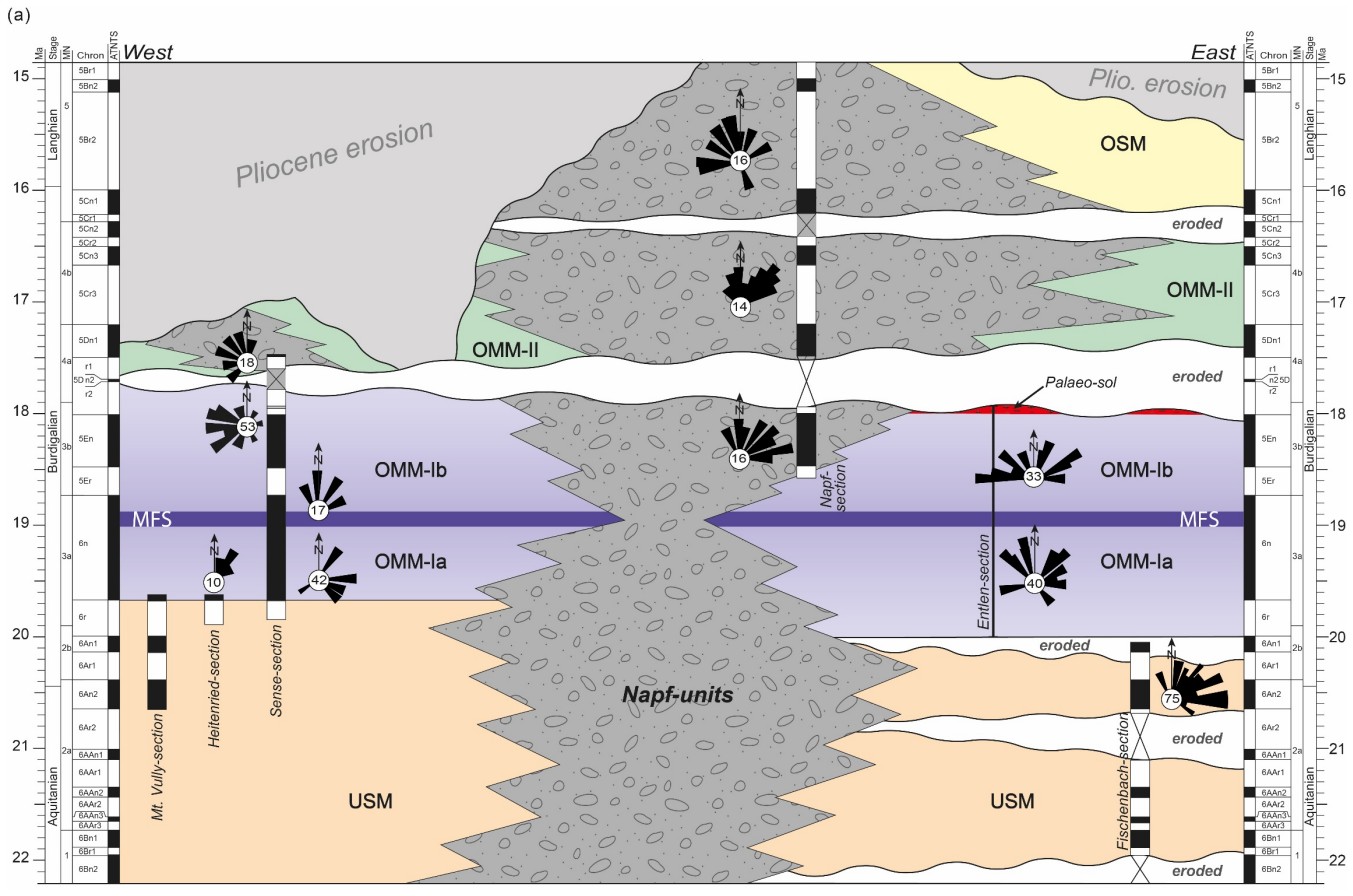

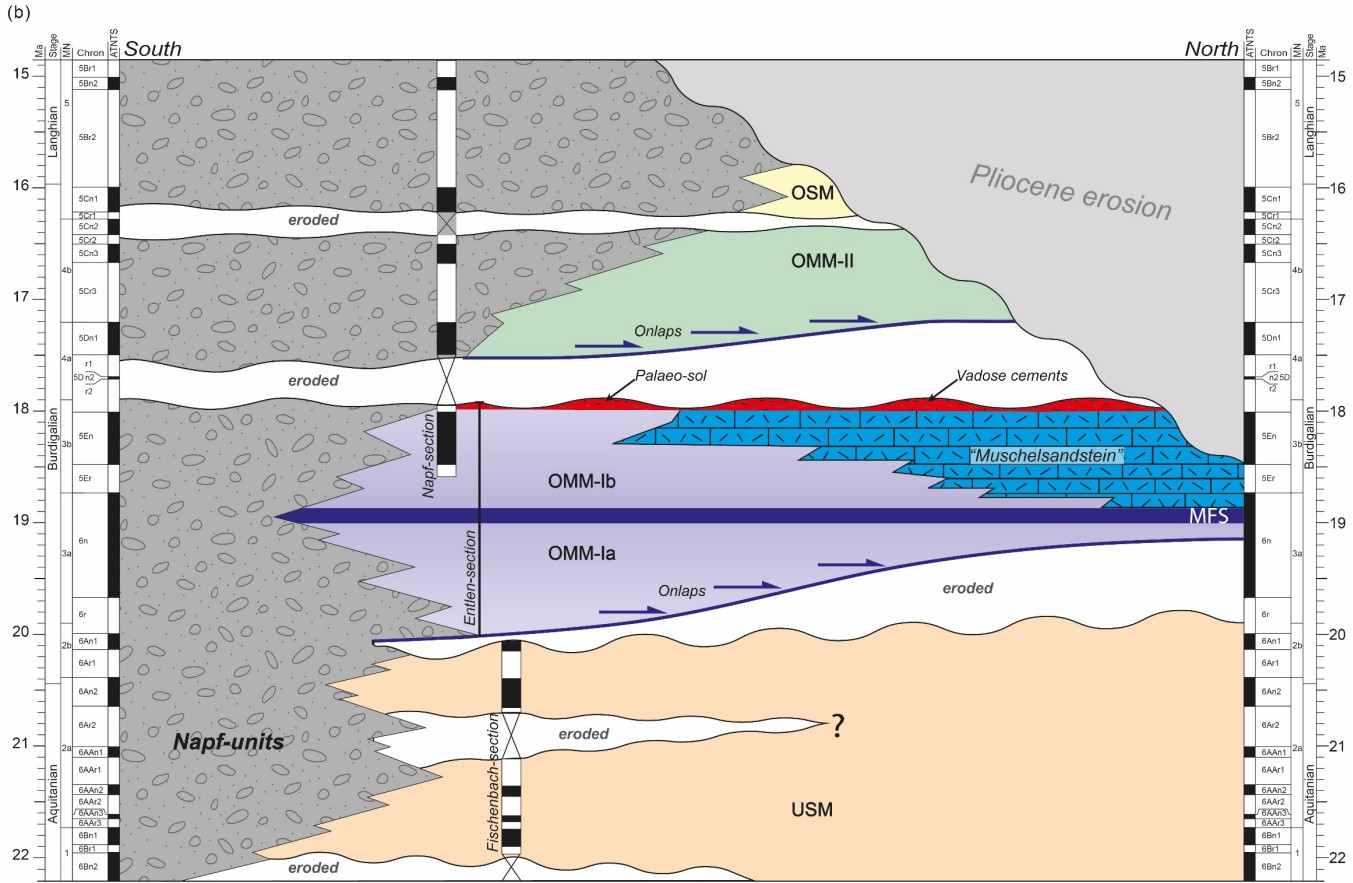

**Figure 5: a)** West-East chronological (Wheeler) diagram of the Molasse sequence at the proximal basin border between Fribourg and Lucerne (Fig. 2a). The following magnetostratigraphic data have been used: Mt. Vully, Heitenried and Sense (Strunck and Matter, 2002), and Napf and Fischenbach (Schlunegger et al., 1996). Palaeo-transport directions from Heitenried and the upper part of the Sense section are taken from Strunck and Matter (2002). Note, that the Entlen section is not calibrated with magnetostratigraphic data but has been adjusted using regional information (see text for further details and Fig. 3b for synthetic sections of the region). MFS = Maximum-flooding surface. Note that Pliocene erosion removed most of the OMM-II record in western Switzerland. We infer marine conditions in the western Swiss Molasse basin during OMM-II times because: (i) marine conditions were present east of the Napf-units, and (ii) material transport occurred towards the West, which implies that marine conditions were also present west of the Napf-megafan at that time as confirmed by mapping (e.g., Wanner et al., 2019); **b)** North-South chronological (Wheeler) diagram of the Molasse sequence between Entlen (site 13) and Madiswil (site 14, both on Fig. 2a). See text for further details. The onlaps (blue arrows) are based on interpretations from seismo-stratigraphic data (Schlunegger et al., 1997). MFS = Maximum-flooding surface.

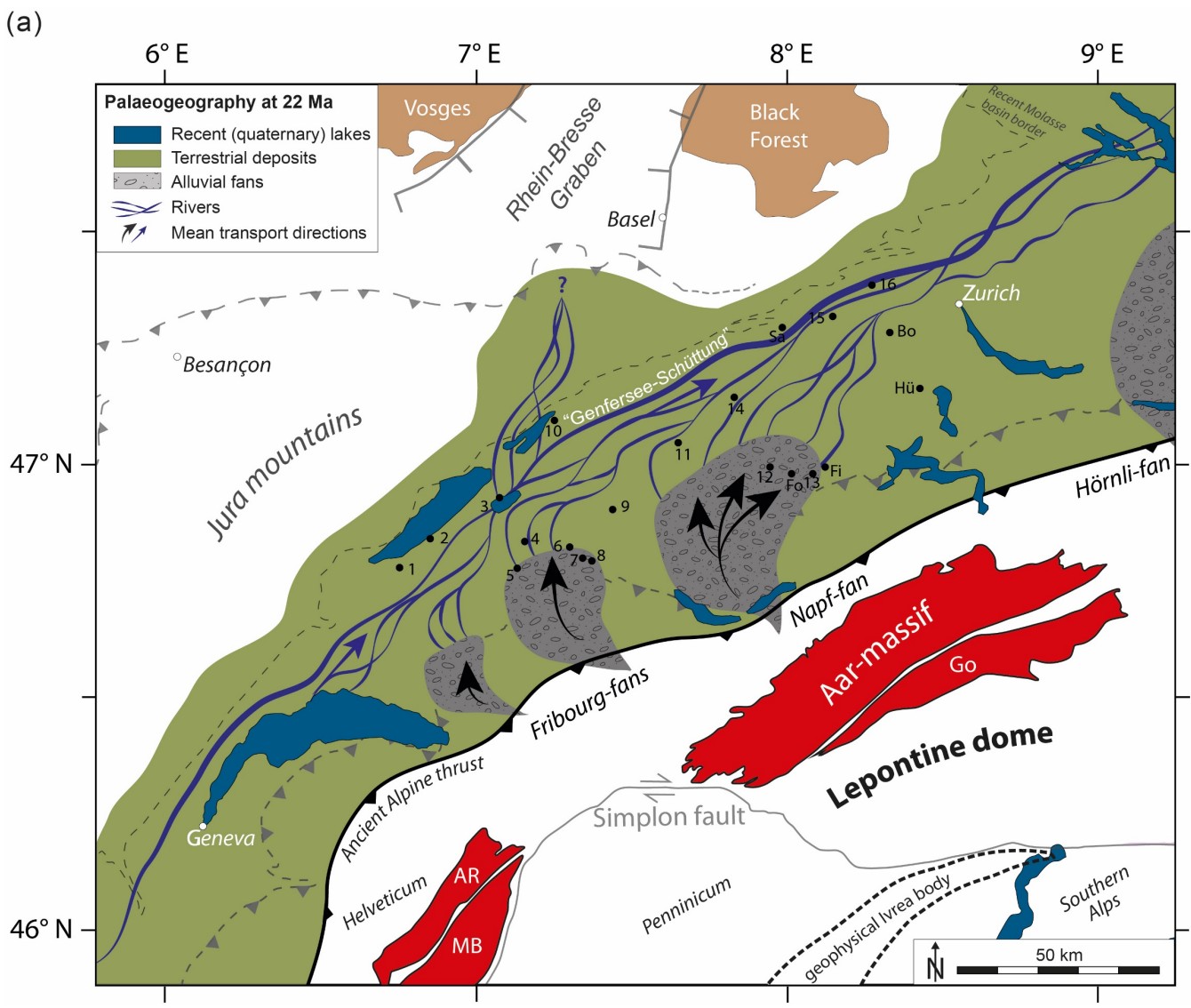

(a)

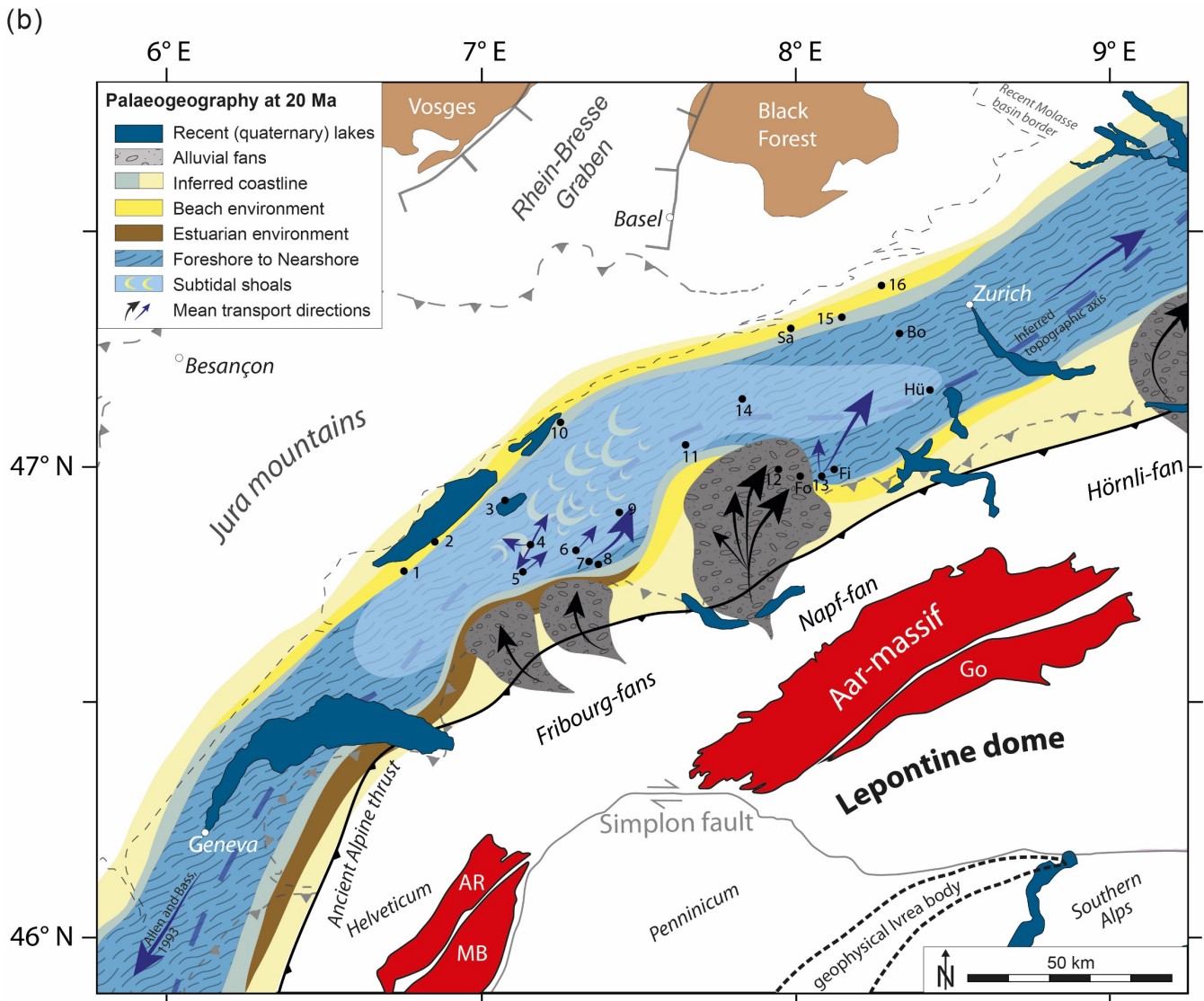

(b)

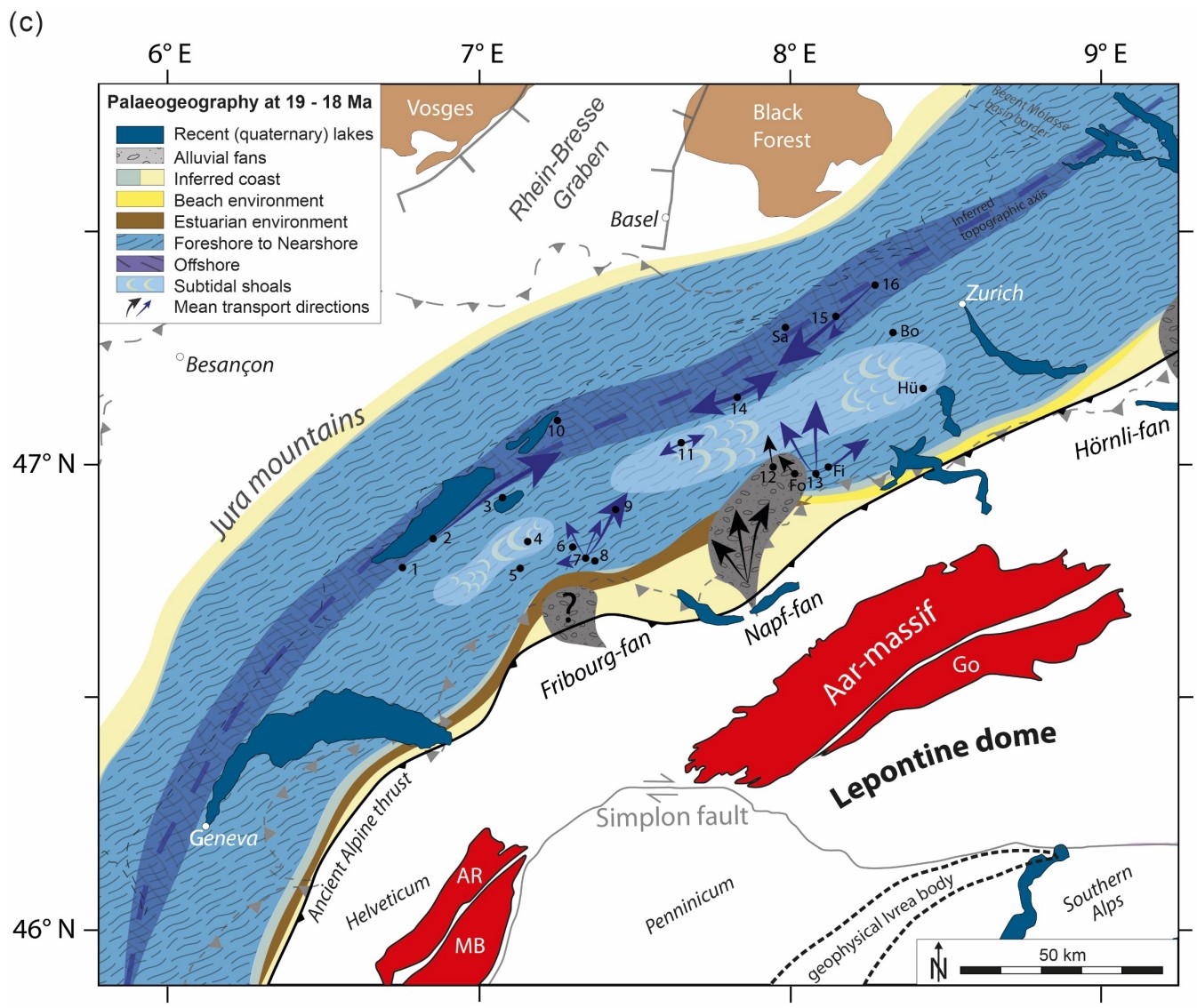

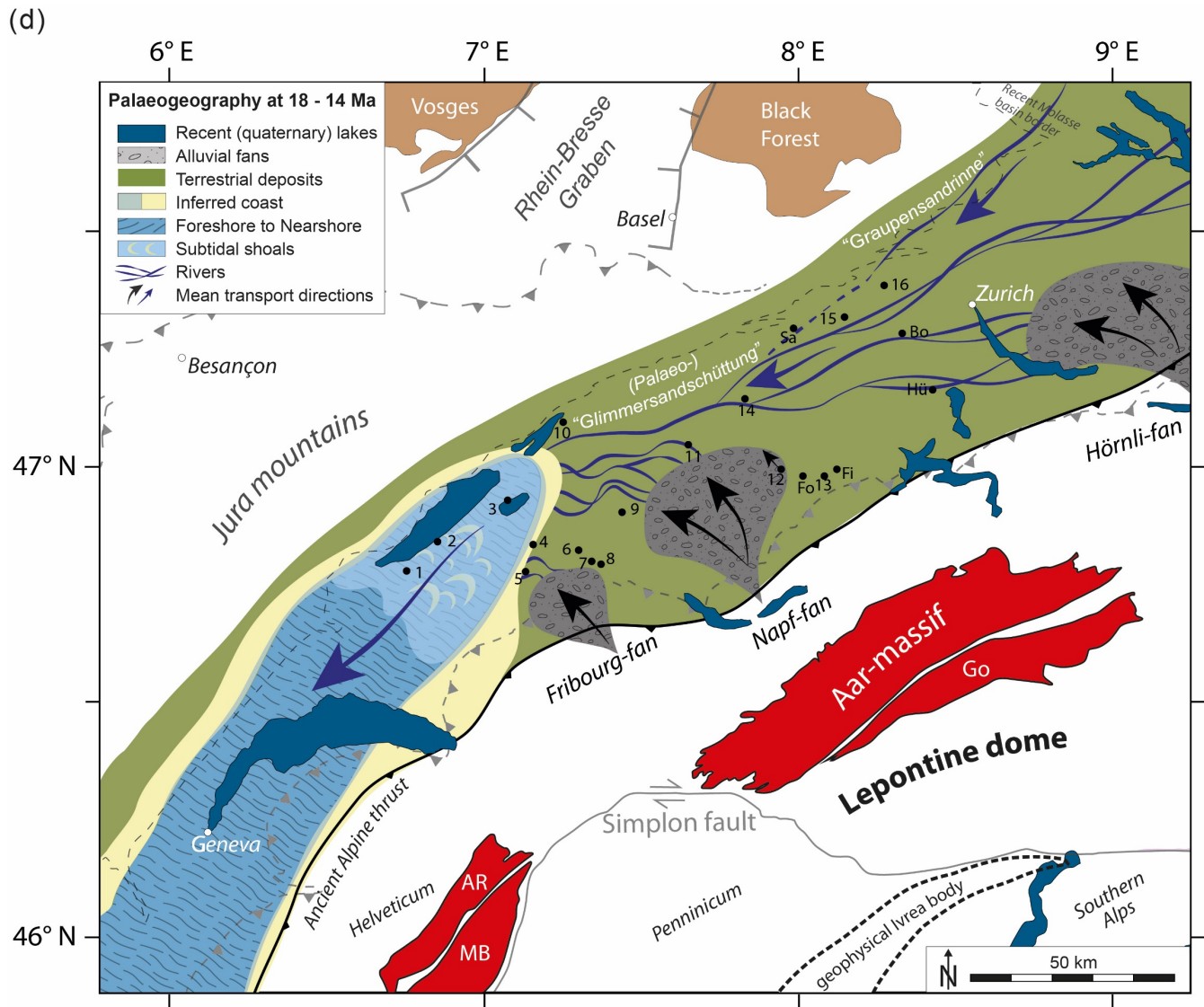

**Figure 6:** Palaeogeographical reconstructions of the Molasse basin at different stages: **a)** USM (c. 22 Ma), **b)** OMM-Ia (c. 20 Ma), **c)** OMM-Ib (c. 19-18 Ma) and **d)** OMM-II to OSM (c. 18 df – 14 Ma) modified after Kuhlemann and Kempf (2002) and based on own observations. Note that all maps show present-day lithotectonic units within the Alps and the Jura mountains for orientation purposes (dashed lines and grey-coloured lines). We acknowledge that the positions of these and the surface patterns (such as lakes) were different during deposition of the Molasse deposits. The location of the palaeo-thrust fronts (thick lines) are adapted from Kuhlemann and Kempf (2002). Black dots mark study sites for orientation purposes. Please refer to Fig. 1 for the complete legend.

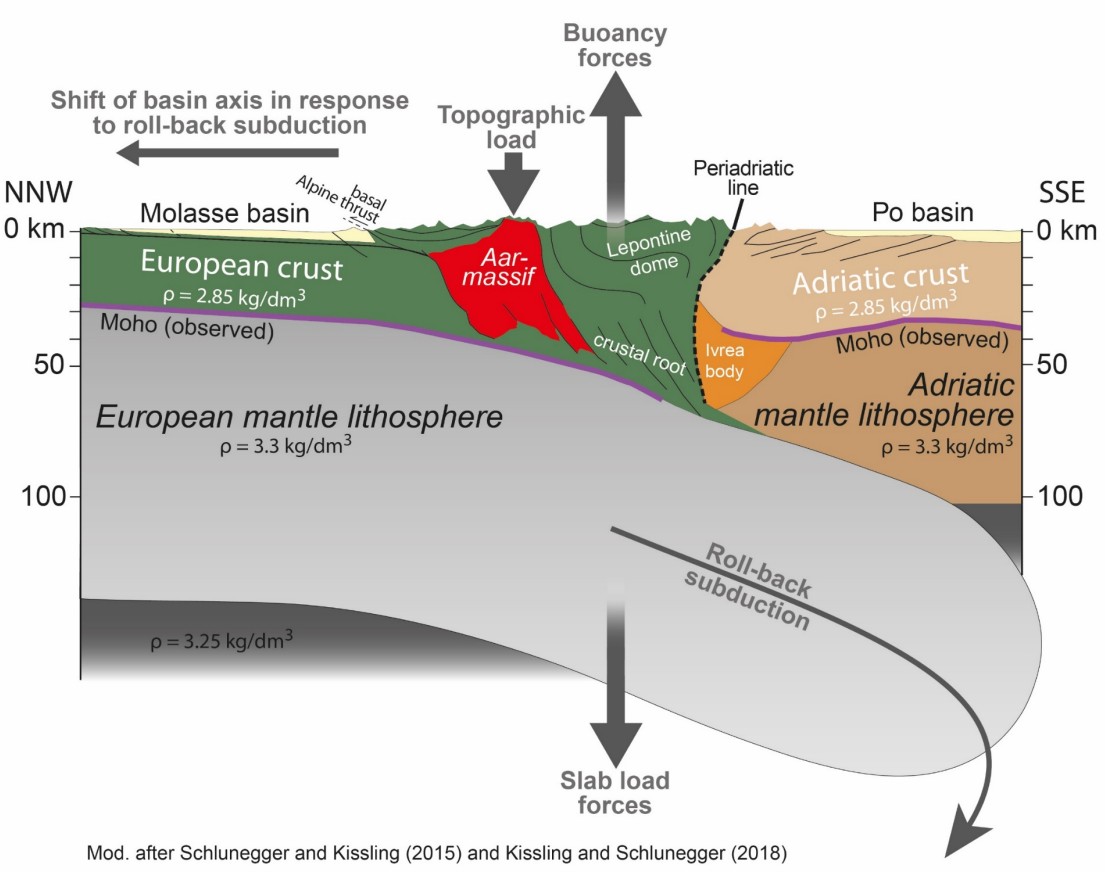

**Figure 7:** Simplified geological-geophysical model of the Alpine orogen for the time between 20 – 18 Ma, showing the most important geodynamic forces that might have shaped the Molasse basin and induced the transgression. Modified after Schlunegger and Kissling (2015) and Kissling and Schlunegger (2018).

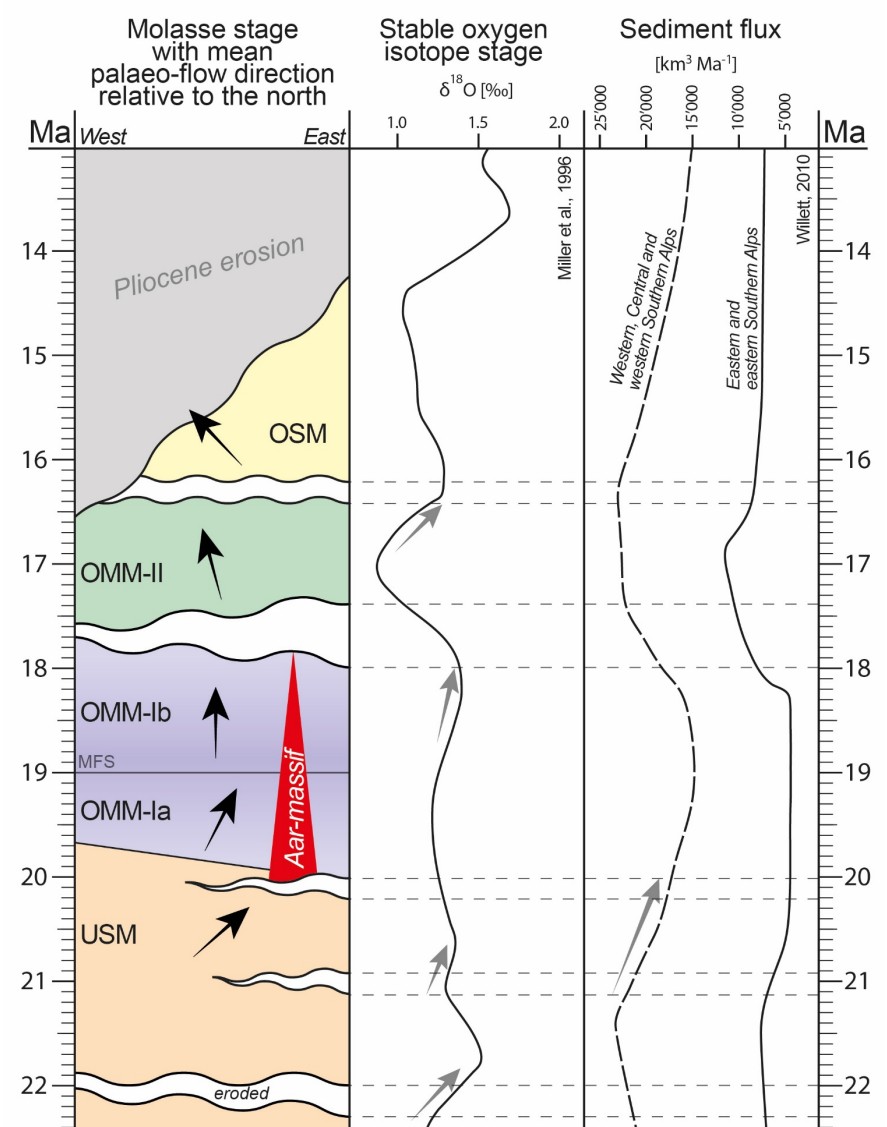

**Figure 8:** Molasse stages (USM, OMM and OSM) with mean palaeo-transport directions (black arrows), hiatuses plotted against stable oxygen isotope stages (Miller et al., 1996) and sediment flux (Kuhlemann, 2000; Willet, 2010). The red triangle demarcates the onset of delamination and rapid exhumation of the Aar-massif (Herwegh et al., 2017). Grey arrows demarcate decreases in sediment flux and falls in the eustatic sea level possibly contributing to the related hiatuses. MFS = Maximum-flooding surface.