# Peer review of "Tectonic processes, variations in sediment flux and eustatic sea level recorded by the 20 Ma-old Burdigalian transgression in the Swiss Molasse basin"

_Solid Earth, 2019_

## Referee Comment (RC1) · Kei Ogata (Referee) · 14 Mar 2019

Dear Editor,

The paper represent a broadband reappraisal of the Alpine Molasse basin through the implementation of different stratigraphic techniques in order to illustrate different scenarios from the lithospheric to the basin scale. Although the large amount of processed material, bibliographic review and some interesting considerations, the overall manuscript reads quite poorly and the overall message struggle to pass through. The

main issues are: 1) unclear subdivision between original data and literature, 2) mixed results-interpretations, 3) poor logic organisation and structure, 4) too long and complicated sentencing and paragraphing, 5) poor graphical support, 6) unclear link between presented data and interpretations, and 7) unbalanced manuscript-supplementary material. These and several other points are detailed in annotated pdf attached.

Sincerely, Kei Ogata

Please also note the supplement to this comment:
https://www.solid-earth-discuss.net/se-2019-27/se-2019-27-RC1-supplement.zip

---

## Author Comment (AC1) · 15 Mar 2019

Dear Editor, Dear Reviewer,

We greatly appreciate the critical, constructive and detailed comments on our manuscript which requires a major reorganisation. Indeed, the comment mainly addresses editorial issues, in the sense that the organisation of the paper is not satisfying. The reviewer suggest numerous points in the annotated manuscript of how the article can be restructured. Apparently, the proposed structure leads to a paper ar-

chitecture which builds on methods, results, interpretation and discussion (the current version is indeed different). We will fully consider this recommendation upon revising our manuscript.

Our contribution is a detailed sedimentological analysis of 5 sites distributed within the Upper Marine Molasse (OMM) in the central north Alpine foreland basin between Lucerne and Fribourg. We used this dataset to reconstruct the transgression and sedimentation pattern across the basin, from which we extract signals related to eustatic, sediment supply and tectonic controls. However, such an analysis also requires a chronological framework of the analysed deposits. Therefore, as a second contribution of our work, we compiled published biostratigraphic and magnetostratigraphic datasets, which resulted in a revised chronological framework of the Molasse deposits in the study area. We acknowledge, that our refined chronological framework mainly results from the discussion of compiled data. Therefore, we consider shifting the synthesis of the magnetostratigraphic and biostratigraphic data to the discussion section (first chapter). The following sections will build on these data and comprise the discussion of the palaeogeographic maps, which in turn base on the sedimentological analysis presented in the result and interpretation sections. As a final part, we will discuss the underlying controls associated with the Burdigalian transgression. We will thus restructure the paper as proposed by the reviewer.

We will also include the table with the sedimentological lithofacies (supplement) in the full text, as suggested. The result section will thus comprise the presentation of the sedimentological observations together with new photos from the field. The reviewer claims, that the shift of the table with the lithofacies assemblages to the main manuscript will allow to shorten the text. We agree on this and will proceed accordingly. Finally, we will certainly improve the link between the supplementary information and the main text, and we will rephrase redundant statements that are outlined by the reviewer in the annotated manuscript.

In summary, we consider the requested changes as justified and fully feasible, and

we are happy and ready to proceed accordingly after we have received the interactive comments of the additional reviewer.

We would like to thank the editor for handling our manuscript and the reviewer for his time he invested in our work.

Sincerely, Philippos Garefalakis & Fritz Schlunegger

---

## Referee Comment (RC2) · Kenneth Eriksson (Referee) · 12 Jul 2019

The authors have integrated a large data base consisting of previous studies and new observations to discriminate the effects of tectonics, eustatic sea level changes and variations in sediment flux in explaining the Upper Marine Molasse in the Swiss Alps. Discriminating between these controls in understanding the stratigraphic record has long been a subject of discussion amongst stratigraphers and sedimentologists and the authors are to be complimented on their contribution to this to this ongoing debate.

[Figure]

The paper consists of 3 main sections, Chronology, Sedimentology and Controls. The first and third sections are well argued but the sedimentology sections requires major revision including drastic shortening and reference to modern and ancient analogs in support of the conclusions of depositional environments. Such references are surprisingly lacking but are essential to presenting convincing interpretations. Also, the sedimentology section contains numerous examples of interpretations within the descriptive sections and vice versa. In its present form, the paper contains too much sedimentological detail that detracts from the overall message of the paper. I suggest reducing the sedimentological descriptions and interpretations by at least 50% in this paper and to prepare a separate paper that focusses on the sedimentology. The parts of the sedimentological analysis that are germane to this paper the recognition of shoreline and offshore subtidal sand shoals whereas the other details are not necessary for this paper.

Please also note the supplement to this comment:
https://www.solid-earth-discuss.net/se-2019-27/se-2019-27-RC2-supplement.zip

---

## Referee Comment (RC3) · Anonymous Referee #3 · 16 Jul 2019

Review of paper by Garefalakis and Schlunegger entitled Deciphering tectonic,eustatic and surface controls on the Burdigalian transgression recorded in the Upper Marine Molasse in Switzerland

General comments In this manuscript authors used new sedimentological and existing geological and geophysical data to assess tectonic, eustatic and surface controls on the Burdigalian transgression in the Molasse Basin. Even through most of the data and ideas appear interesting and important; there are some fairly significant items that need modification prior to publication. The comments provided below will require ma-

[Figure]

jor revision of the manuscript. Manuscript structure needs reorganization. 1) There is no clear separation between existing data and author's own original data. Result and Discussion section include background information that should be presented earlier in Geological setting (section 2); 2) a clear separation of observations and interpretations is missing in Result section; 3) Scientific methods and workflow are not clearly presented; 4) Headings are not informative; 5) Remove of unnecessary repetitions would cut text significantly. Manuscript needs clearer explanation of the links between their own data and conclusions. Authors often jump into conclusions without showing clear link to either their own field data or literature. First, key sedimentary features observed during this study, that could be used to decipher tectonic, eustasy and surface controls, should be better described. Most of important observations in that respect are mentioned for the first time in Discussion section. Second, there is a confusing separation of the processes operating at the lithospheric - and crustal - scale like they are not interacting at all. These processes are poorly defined in the paper and their links to author's field data are not clear. This needs to be improved prior to publication. Detailed examples of problem areas in the text are given below and in the attached pdf.

Specific comments Introduction. Opening paragraph of the introduction needs to be focused. Motivation to undertake this study is not clear. What is so controversial about Molasse Basin, i.e. Burdigalian transgression to be further studied? It is not clear what is considered by term surface controls? Settings. Section on geological background should be extended. I recommend starting by adding information on formation and geodynamic evolution of the Alps. Special attention should be given to Aar Massif, Simplon detachment and Lepontine dome (i.e. kinematic, geometry, evolution, lithology of the units involved in faulting etc.) that are in the further text marked as important controls on deposition in Molasse Basin. Section 2.2 - Molasse Basin - state of the art, particularly studied Upper Miocene Unit, is poorly defined, most of important background information appears in Results and Discussion. Methods. I suggest to explain and list all the methods used in your study. Also, list the methods in the same order that they will appear in the results. Avoid general sentences with vague point. Explain

[Figure]

which sequence stratigraphic approach was used. Heading 3.1 is misleading because this section mentions stratigraphic methods as well. Subchapter 3.2. is not needed. It is difficult to distinguish background data and methods. It looks like reinterpretation of the literature data. Results. I recommend to start with describing your data and avoid mixing it with interpretation in this section. As written - the text is currently hard to follow. Moreover, section 4.1. includes background information that should be part of Geological setting section and Discussion. In the subsection 4.2. please systematically lay out your observations. I suggest grouping already defined lithofacies types into facies associations that are typical for particular depositional environment. This should be followed by definition of stratigraphic sequences that can be further link to suggested controls. Furthermore, this should be associated with illustrations such as your own logs or field photos that show characteristic sedimentary packages and/or stratigraphic surfaces. By doing so, you would be able to follow vertical and lateral transitions and interpret them in the light of tectonic and eustatic controls on the basin evolution. Very important in the section 4.2 Interpretation part – references are completely missing! Discussion. Section 5 should be moved to Discussion. I recommend starting this section with the ideas on basin evolution based on your own findings. Some basin features e.g. backstepping of the alluvial mega fans are described for the first time in this section. Furthermore, it is not clear which mechanism controlled it.

Figure comments Minor comments are included in attached pdf. Figure 5. How did you construct mean water depth curve? In some instances, you have contradiction between your sedimentological and paleo-depth data. Please revise curve.

Please also note the supplement to this comment:
https://www.solid-earth-discuss.net/se-2019-27/se-2019-27-RC3-supplement.pdf

**Supplement:**

[revised manuscript text omitted]

---

## Referee Report (RR1)

[referee-annotated manuscript omitted]

---

## Author Response (AR3)

Dear Editor,
Dear Reviewer,

Thank you very much for organizing a further set of comments and suggestions, which we considered very helpful for clarifying the statements of our paper. We considered all of them and have updated the paper accordingly. As a consequence, we modified the title of the manuscript such as that it better expresses what the paper is about. As outlined below, you will find the list of comments, which we have addressed in the revised manuscript point by point.

**Anonymous Referee #3**
*General comments*
The manuscript has been significantly improved, however there are still some items that need to be improved prior to publication. Detailed examples of problematic areas in the text are given below and in the attached pdf.

We thank the Reviewer for the suggestions and constructive criticisms related to our work. We have addressed all points and updated and improved the paper accordingly. We realized that the suggestions resulted in a clarification of our major points, which we see as added value.

Manuscript text needs to be improved. Text often includes long and complicated sentences. Furthermore, the writing style in the manuscript is not balanced.

The Reviewer outlined the complicated sentences in the attached pdf and proposed a way of how they can be changed; we followed these recommendations.

Some chapters such as methods and results are written much better than others. There are still some repetitions in the text.

The repetitions have been removed where possible. We realized that some few statements needed to be brought up again, else the argumentation was difficult to follow in some places.

Discussion needs to be clearer and supported by data and references.

We improved the discussion section accordingly.

1) Hypothesis stated in the text are often not clarified and adequately explain/supported;

This has been corrected.

2) Titles do not correlated with the text in subchapters (e.g., Controls on the establishment of a wave-dominated coast in the east and tidal records in the west, page 20);

We addressed and corrected this issue. The headings are now more informative and correlate with the text.

3) Mixing of poorly or non-related features in the subchapters.

We have updated the text such as that we particularly address this point. Subchapters and headings now clearly express what the content is about.

The predominance of tectonics over eustatic sea level effects on marine transgression is poorly explained. In the text, the effect of sea level rise was not discussed, only sea level drop. Fig. 8 shows correlation between sea level rise and OMMI and OMMII. How does tectonics explain westward progression of marine transgression at the base of OMM? How does exhumation of Aar affect lateral variations in wave - vs tide - dominated deposition, but does not affect along-strike coeval appearance of MFS (Fig. 5a)?

We paid attention to this point and explained the role of a rising sea level more carefully, please see sections 6.3.3. and 6.3.5.

*Specific comments*

Abstract
Very difficult to follow. Goal of the study is missing. Explanation of main finding is vague and confusing. Suggestion > explain the main findings in the same order as listed in the line 17.

We have modified the abstract and clarified the major findings.

Geological settings
The chapter 2.1 seems to be more informative compared to previous manuscript version, however it is difficult to follow. The description of orogen architecture and evolution are mixed. The authors often jump from description of crustal features to lithospheric features and suddenly back again to crust. Process of delamination was described in very confusing way.

We improved this chapter and have clarified these points following the suggestions by the Reviewer that were formulated in the annotated pdf.

Discussion: 6.2. Evolution of the Molasse Basin
It seems that this study mainly confirmed results derived from previous studies e.g. Pfiffner et al., 2002; Kuhlemann and Kempf, 2002 and references therein. If so, than it should be clearly stated at the beginning and this chapter should be shortened. Otherwise, new findings and potential contradictions resulted by this research should be highlighted.

We clarified the separation between confirmed results and own new findings at the end of this chapter (please see p. 18, lines 33 ff).

6.3. Controls on the marine transgression of the OMM
This chapter should start with discussion on factors that control transgression i.e. processes that leads to creation of accommodation space more rapidly than it is consumed by sediment supply. For example sea level or processes that control sediment supply. Reversal of the drainage direction might be just a consequence of the process that has not much to do with transgression. Avoid explaining processes that are not related in the same chapter or explain how they relate (e.g., How are the drainage reversal and basin widening related?, form the text below seems that are not related).

We agree on these points and realise that our statements have not been sufficiently clear. We have thus modified the corresponding paragraphs to better explain our interpretation, and we have adjusted the heading of the subchapters for clarification purposes.

6.3.1 (perhaps) Possible controls on the reversal.

In this subchapter should be listed other hypothesis related to the controls on the reversal (for example rise of Amstetten Swell) and reasons why author prefers Pfiffner's hypothesis over others. However, It is not clear the link between text that follows and Pfiffner's hypothesis.

This has been adapted and improved (please see lines 14 – 18 on p. 19). We have discussed why the hypothesis by Pfiffner et al. (2002) bears a valid point, but also why a view at a larger scale is required.

The hypothesis that slab break off/delamination in the Eastern Alps controls drainage reversal is very interesting and novel (although it is vaguely proposed by Handy et al., 2015). Thereby, this should be clearly stated as author's hypothesis (new contribution to the current knowledge on Molasse basin evolution) and argued why it is favored over other hypothesis. Very important: explanation of independent geological evidences that support this hypothesis are missing. Potential triggering mechanisms of lateral variations in the slab load are poorly explained. For example what would be the wavelength of this process (whole basin and hinterland, part of the basin?)?Suggestion > please avoid citing papers that are not related to the subject, for example Mey et al., 2016, better try with Waschbusch and Royden, 1992 and others.

We acknowledge that we cannot address all these mechanisms at this stage, because this would be a paper of its own. Nevertheless, we changed and improved the manuscript where we have the required information. We also acknowledge in the text that this chapter is still hypothetical at this stage, and we await the results of the ongoing AlpArray project (mentioned at the end of the paper). We basically need more information.

Controls on the establishment of a wave-dominated coast in the east and tidal records in the west How is this related to transgression? It is hard to understand how lithospheric-scale processes (slab-roll back mechanism leading to delamination of upper continental crust) are linked to and potentially triggering wave- vs. tide-dominated deposition without taking into account (discussing) other effects (such as wave vs tide energy, tidal waves from Mediterranean and Paratethys, local climate, river discharge).

The main point is that strong tides are recorded in the basin axis at the distal East and West, but not at the proximal basin border to the east of the Napf. Therefore, differences in surface forces are excluded. We explain that the records also depend on water depth, which appear to have changed in response to the buckling of the underlying plate, driven by the uplift of the Aar-massif. Please see lines 3 on p. 21 to 14 on p. 21.

Regarding the location of the thrust related signal in the basin (page 20, line 20). The Te calculated for the Swiss Alpine foreland varies from 5 to 50 km (Pfiffner et al., 2002 and references therein). Thereby, the location of the expected signal would vary. This might not provide substantial resolution to explain link the location of the bulge and transition from tide- to wave-dominated depositional environment.

We relate these criticisms to the poor explanation of our interpretation and have updated the text accordingly for clarification. Please see lines 18 on p. 21 to 4 on p. 22.

Controls on sediment supply and sea levels variations are poorly defined. For example the subchapter on sea level variations gives impression that transgression (with MFS between OMMIa and OMMIb) is controlled by eustatic sea level rise. Sea level rise can induce widening of the basin.

We have discussed the possible controls of a rising sea level more carefully. In fact, this could explain the establishment of the OMM-II, but the rise is probably not sufficient to allow the transgression of the OMM-I. We see the OMM-I transgression as the consequence of a reduced sediment flux and an increase in the tectonically controlled accommodation space. Eustatic sea level highstands could have amplified these effects. This is explained more carefully in section 6.3.5.

Summary and Conclusions
It is necessary to clearly highlight main conclusion of this study and avoid repetition. Explanation how did tectonic triggered marine ingression into Molasse Basin.

This has been done.

Dear Editor,
Dear Reviewer,

We would like to thank you for the very constructive reviews, which we have received with much appreciation. The comments and suggestions by the referees contributed to greatly improve the organization and the science of our paper.

The major changes include

- A better focus of the introduction, thereby addressing the aim of the manuscript in a clearer way.
- An expansion of the state of knowledge, explaining the architecture and evolution of the Alps and the Molasse basin, and an overview of published work on the Upper Marine Molasse
- A clear, but concise, focus on the sedimentology of the Upper Marine Molasse, which is our major contribution
- A separation between methods, results and interpretation of the sedimentological data
- A presentation of references to published sedimentological work within an overview table
- A plate with photos from the field in the supplement
- A re-organization of the discussion. We have reframed the re-assessment of the chronological framework as a first chapter of the discussion, followed by a discussion of the evolution of the Molasse basin within a geodynamic framework.
- We linked subduction tectonics with (i) the formation of accommodation space in the Molasse basin and with (ii) changes of the drainage network on the surface of the Alps, which could explain the reduction of sediment flux.

We have addressed all other questions and suggestions made by the reviewer. Below, we present a point-by-point response of how we have modified the manuscript.

**Referee #1: Kei Ogata**

Dear Editor,
The paper represent a broadband reappraisal of the Alpine Molasse basin through the implementation of different stratigraphic techniques in order to illustrate different scenarios from the lithospheric to the basin scale. Although the large amount of pro- cessed material, bibliographic review and some interesting considerations, the overall manuscript reads quite poorly and the overall message struggle to pass through. The main issues are: 1) unclear subdivision between original data and literature, 2) mixed results-interpretations, 3) poor logic organisation and structure, 4) too long and compli- cated sentencing and paragraphing, 5) poor graphical support, 6) unclear link between presented data and interpretations, and 7) unbalanced manuscript-supplementary ma- terial. These and several other points are detailed in annotated pdf attached.
Sincerely, Kei Ogata

***Our response:***
We have fully restructured the article to comply with these requests:

1) Original and literature data has been separated into distinct chapters following the recommendations of reviewer 3. We also have largely expanded the setting chapter where we outline the state of current knowledge about the Alps, the Molasse and the related evolutionary processes. These built the basis for the discussion presented in chapter 6.

2) Results and Interpretations are now presented in two different chapters. Pleases note that all references in this context are presented in tables 1 to 5, which are now fully integrated in this text. We decided to proceed in this way because the text is easier to read.

3) We have fully reconsidered the organization and structure of the paper according to the current state of the art. Introduction is now followed by an extended local setting chapter, an extended method section, followed by results, interpretation, discussion and conclusion. Here we mainly followed the recommendations by reviewer 3.

4) We have corrected and shortened the text particularly in the middle part where we present the sedimentological observations. Nevertheless, we decided not to cut the level of information and thus shifted more details into tables 1 to 5, which are now fully integrated into the text. This was also recommended by reviewer Ogata. Reviewer Eriksson provided useful comments in this regards, which we fully acknowledged and took into account.

5) We included two more figures in order to provide more information to follow the text. In addition, based on the detailed comments in the annotated documents, reviewer Ogata required that the figures should be better integrated into the text. We have seriously considered this point and updated the text accordingly. In this context, we greatly appreciate the suggestions by reviewer 3 in his annotated text.

6) Please see our response above. In addition, we have made distinct links between our sedimentological observations and interpretations, and the discussion. This was also required by reviewer 3, who made clear suggestions of how to proceed. We adapted all recommendations by reviewer 3 and updated the text accordingly. This resulted in re-structuration of the discussion.

7) We shifted the tables with sedimentological details to the main text as required by reviewer Ogata. We have improved the referencing between the main text and additional information presented in the supplementary file.

In addition, we followed all recommendations presented in the annotated file. These mainly concern editorial requests.

**Referee #2 – Kenneth Eriksson**

*General comments:*
The authors have integrated a large data base consisting of previous studies and new observations to discriminate the effects of tectonics, eustatic sea level changes and variations in sediment flux in explaining the Upper Marine Molasse in the Swiss Alps. Discriminating between these controls in understanding the stratigraphic record has long been a subject of discussion amongst stratigraphers and sedimentologists and the authors are to be complimented on their contribution to this to this ongoing debate.

The paper consists of 3 main sections, Chronology, Sedimentology and Controls. The first and third sections are well argued but the sedimentology sections requires major revision including drastic shortening and reference to modern and ancient analogs in support of the conclusions of depositional environments. Such references are surprisingly lacking but are essential to presenting convincing interpretations. Also, the sedimentology section contains numerous examples of interpretations within the descriptive sections and vice versa.

In its present form, the paper contains too much sedimentological detail that detracts from the overall message of the paper. I suggest reducing the sedimentological descriptions and interpretations by at least 50% in this paper and to prepare a separate paper that focusses on the sedimentology. The parts of the sedimentological analysis that are germane to this paper the recognition of shoreline and offshore subtidal sand shoals whereas the other details are not necessary for this paper.

*Our response:*
Thank you for your detailed review and the constructive comments. The sedimentological descriptions and interpretations have been split into two chapters (4. Results and 5. Sedimentological interpretation) and have been substantially shortened. We now present key information only, which will be crucial for following the discussion about tectonics, sediment flux and eustacy as possible controls on the Burdigalian transgression. For the sake of completeness, however, we list all sedimentological details in a table together with the references to previously published work on this topic.
We also include a plate with photos from the field as required by reviewer Ogata.

*Referee:*
*Specific comments:*
1. Use of the term "surface controls" in the title and throughout the paper is vague and confusing. Both eustasy and sediment flux are surface controls as noted by the authors so why not just specify eustasy and sediment flux and do away with "surface"?

*Our response:*
This has been improved. We specified the term surface controls, which include changes in sediment flux and shifts in the eustatic sea level. These processes have been placed in a geodynamic framework together with deep crustal processes. These mechanisms encompass tectonic changes at the slab scale in the mantle lithosphere, and related to these mechanisms crustal-scale processes at a more regional scale.

*Referee:*
2. I was not able to access the Table or Appendix but it seems to me that the 2 seismic sections should be included as a figure in the paper because they are referred to in many parts of the text.

*Our response:*

We apologize for this inconvenience. In the supplement file the reader finds a detailed description on how we calculated the palaeo-bathymetrical conditions from wave ripple marks and from the set-thickness of sedimentary bedforms. Furthermore, also in the supplement file, we marked the relevant part of the seismic line BEAGBE.N780025 in Fig. S4. The seismic line 8307 has been fully published in Schlunegger et al., 1997a (see revised manuscript), so we have not reproduced this section.

The original table S2 of the supplement file has now been included in the main text and split into 5 individual tables where each contains the abbreviations of the facies assemblages, the description of the bedforms and the resulting depositional setting together with a list of references.

*Referee:*

*Technical suggestions:*

1. The attached document contains numerous grammatical and editorial suggestions and comments on both the text and figures for the authors to consider in their revision.

***Our response:***

We have considered all suggestions upon revising our paper.

***Referee:***

2. As part of my review, I have prepared a document of revised figure captions, which is attached for the authors' consideration.

***Our response:***

We greatly acknowledge the careful and detailed work and have considered all points upon revising our paper. Please note, however, that we have rephrased most of the sections to comply with the comments of reviewer 1 and 3.

**Anonymous Referee #3**

*General comments:*
In this manuscript authors used new sedimentological and existing geological and geophysical data to assess tectonic, eustatic and surface controls on the Burdigalian transgression in the Molasse Basin. Even through most of the data and ideas appear interesting and important; there are some fairly significant items that need modification prior to publication. The comments provided below will require major revision of the manuscript. Manuscript structure needs reorganization.
1) There is no clear separation between existing data and author's own original data. Result and Discussion section include background information that should be presented earlier in Geological setting (section 2).

*Our response:*
This has been corrected. We improved the manuscript accordingly and provided more information on previously published geologic, chronologic and geodynamic data at the beginning of the paper. This concerns both the Alps and the Molasse basin. In this context, we have added new chapters and thus expanded the local setting significantly. We also restructured the paper such as that description of data and interpretation is clearly separated. The re-assessment of the Molasse chronology is shifted to the discussion section as a first chapter, as this could be considered as a discussion.

*Referee:*
2) a clear separation of observations and interpretations is missing in Result section;

*Our response:*
This has been done. Methods, Results and Interpretation are presented in separate chapters. We have also significantly expanded the Methods section to provide more information about how we have proceeded in the field, how we have measured paleo-flow directions and how we have collected information on the facies patterns.

*Referee:*
3) Scientific methods and workflow are not clearly presented;

*Our response:*
This has been improved. The various sedimentological methods include logging, paleo-flow measurements, estimates of paleo-water depths and mapping, and taking field notes. We have provided more details on these aspects. The results are organized according to (i) analyzed sections (e.g., Entlen section, Sense section etc), and (ii) the methods applied in the field. Such a re-organization was also requested by reviewer 1. The interpretation follows the same structure. The discussion starts with a re-appraisal of the chronological framework (same as requested by reviewer 1), followed by a discussion of the basin evolution and a possible relationship to tectonic and eustacy controls. In this regard, we combined the erosional flux scenario with tectonic processes (i.e. tectonic processes that control significant shifts in the drainage network with a negative feedback on sediment flux). We have discussed these points more clearly as requested, and we have also refined the workflow, which appears now much clearer to us. Thanks for pointing this out!

*Referee:*
4) Headings are not informative;

*Our response:*

We have changed nearly all headings such as that they are more informative.

***Referee:***
5) Remove of unnecessary repetitions would cut text significantly. Manuscript needs clearer explanation of the links between their own data and conclusions. Authors often jump into conclusions without showing clear link to either their own field data or literature. First, key sedimentary features observed during this study, that could be used to decipher tectonic, eustasy and surface controls, should be better described. Most of important observations in that respect are mentioned for the first time in Discussion section.

***Our response:***
This has been improved. We carefully streamlined the entire text and shifted published information on the architecture and evolution of the Alps and the Molasse basin to the setting section. We paid special attention on making explicit links between interpretation and observations in all sections of the discussion, and we updated and improved the text accordingly. We additionally paid special attention on carefully addressing all comments that in the annotated manuscript.

***Referee:***
Second, there is a confusing separation of the processes operating at the lithospheric - and crustal - scale like they are not interacting at all. These processes are poorly defined in the paper and their links to author's field data are not clear. This needs to be improved prior to publication. Detailed examples of problem areas in the text are given below and in the attached pdf.

***Our response:***
This has been solved. We presented the general knowledge about the processes on the surface of the Alps and at the crustal levels in the setting section. We then placed the surface and tectonic processes in one geodynamic framework, and we rephrased the discussion section accordingly such as that the reader gets a view of how lithospheric and surface processes were closely linked, which finally resulted in the transgression of the Upper Marine Molasse. We also considered all individual annotations in the attached PDF (see also response above).

***Referee:***
Specific comments Introduction. Opening paragraph of the introduction needs to be focused. Motivation to undertake this study is not clear. What is so controversial about Molasse Basin, i.e. Burdigalian transgression to be further studied? It is not clear what is considered by term surface controls?

***Our response:***
This has been improved. We specify the non-solved problems regarding the Burdigalian transgression and then outlined more clearly the aim of our contribution.

***Referee:***
Settings. Section on geological background should be extended. I recommend starting by adding information on formation and geodynamic evolution of the Alps. Special attention should be given to Aar Massif, Simplon detachment and Lepontine dome (i.e. kinematic, geometry, evolution, lithology of the units involved in faulting etc.) that are in the further text marked as important controls on deposition in Molasse Basin.

***Our response:***
Done. We organized the setting such as that the architecture of the Alps is being presented first, followed by the geodynamic development. We did the same for the Molasse Basin. Please also

see response above.

*Referee:*
Section 2.2 - Molasse Basin - state of the art, particularly studied Upper Miocene Unit, is poorly defined, most of important back- ground information appears in Results and Discussion.

*Our response:*
We have shifted the relevant information from the discussion to the setting chapter.

*Referee:*
Methods. I suggest to explain and list all the methods used in your study. Also, list the methods in the same order that they will appear in the results. Avoid general sentences with vague point. Explain C2 which sequence stratigraphic approach was used.

*Our response:*
Done. Methods, Results and Interpretation are organized such as that the same line is being preserved throughout the paper. We have expanded the related sections and added additional information on our tasks in different sub-chapters.
We actually identify parts within the analyzed Entlen and Sense sections that possibly record the maximum flooding conditions of the transgression. These were then used as correlation tools across the basin. This has been clarified in the revised manuscript.

*Referee:*
Subchapter 3.2. is not needed. It is difficult to distinguish background data and methods. It looks like reinterpretation of the literature data.

*Our response:*
We have removed this section.

*Referee:*
Results. I recommend to start with describing your data and avoid mixing it with interpretation in this section. As written - the text is currently hard to follow. Moreover, section 4.1. includes background information that should be part of Geological setting section and Discussion.

*Our response:*
Done. Results and Interpretation are now separated in different chapters. In addition, the description of the sedimentological data focuses on those aspects only that will be relevant for the discussion. All other detailed data is summarized in a table to give the reader the full sedimentologic information that can be extracted from the stratigraphic sections. Accordingly, the revised text presents the key information only, which will be used to link the basin stratigraphy to eustacy, sediment flux and tectonics. All additional information on lithofacies is presented in a table together with full references to previously published work.

*Referee:*
In the subsection 4.2. please systematically lay out your observations. I suggest grouping already defined lithofacies types into facies associations that are typical for particular depositional environment. This should be followed by definition of stratigraphic sequences that can be further link to suggested controls. Furthermore, this should be associated with illustrations such as your own logs or field photos that show characteristic sedimentary packages and/or stratigraphic surfaces. By doing so, you would be able to follow vertical and lateral transitions and interpret them in the light of tectonic and eustatic controls on the basin evolution.

***Our response:***

This has been done. The logs are presented on Figures 4a and 4b (summary logs); the details are outlined in a table, and additional photos will be included in the paper as supplement. We mark the location of the maximum flooding surfaces on Figures 4a and 4b.

***Referee:***

Very important in the section 4.2 Interpretation part – references are completely missing!

***Our response:***

They were all presented in the table, which we originally have placed in the supplement. However, we see the necessity to present all information in the main text. We thus re-organized our table, shift it to the main text and add references there. We see this as compromise between a clear acknowledgment of previously published work, and the readability of the text (which can be complicated if there are too many references).

***Referee:***

Discussion. Section 5 should be moved to Discussion. I recommend starting this section with the ideas on basin evolution based on your own findings. Some basin features e.g. backstepping of the alluvial mega fans are described for the first time in this section. Furthermore, it is not clear which mechanism controlled it.

***Our response:***

This has been done. We also paid special attention on the 'backstepping' issue and have presented the argumentation and the underlying observations more clearly. In addition, we have specified the underlying controls on the transgression in the revised manuscript. We see subduction tectonics as the principal driving force, with contributions by the uplift of individual crustal blocks (here the Aar-massif) and tectonic exhumation, all of which are related to the subduction processes. In addition, the reduction of sediment flux was likely to have been controlled by the tectonics as well through reorganizations of the drainage network when the basement blocks became uplifted. Eustatic changes in sea level possibly explain the hiatus. We have carefully modified the manuscript to make these points clearer.

***Referee:***

Figure comments: Minor comments are included in attached pdf. Figure 5. How did you construct mean water depth curve? In some instances, you have contradiction between your sedimentological and paleo-depth data. Please revise curve.

***Our response:***

We considered all figure comments and revised the bathymetry curve as requested.

[revised manuscript text omitted]

(a)

Palaeogeography at 22 Ma

- Recent (quaternary) lakes
- Terrestrial deposits
- Alluvial fans
- Rivers
- Mean transport directions

Vosges

Black Forest

Rhein-Bresse Graben

Basel

Recent Molasse basin border

Zurich

Besançon

Jura mountains

"Genfersee-Schüttung"

Hörnli-fan

Napf-fan

Aar-massif

Go

Lepontine dome

Fribourg-fans

Ancient Alpine thrust

Simplon fault

Geneva

Helveticum

AR

MB

Penninicum

geophysical Ivrea body

Southern Alps

50 km

N

[Figure]

(b)

Palaeogeography at 20 Ma

- Recent (quaternary) lakes
- Alluvial fans
- Inferred coastline
- Beach environment
- Estuarian environment
- Foreshore to Nearshore
- Subtidal shoals
- Mean transport directions

[Figure]

(c)

**Palaeogeography at 19 - 18 Ma**
- Recent (quaternary) lakes
- Alluvial fans
- Inferred coast
- Beach environment
- Estuarian environment
- Foreshore to Nearshore
- Offshore
- Subtidal shoals
- Mean transport directions

Vosges

Rhein-Bresse Graben

Black Forest

Basel

Besançon

Jura mountains

Zurich

15
Sa
Bo
Hü

4
6
8
5

Fo 13 Fi

Hörnli-fan

Napf-fan

Fribourg-fan

Aar-massif
Go

**Lepontine dome**

Geneva

Ancient Alpine thrust

Helveticum

AR

MB

Simplon fault

Penninicum

geophysical Ivrea body

Southern Alps

N

50 km

46° N

47° N

6° E    7° E    8° E    9° E

(d)

[Figure]

(a)

[Figure]

[Figure]

(b)

Palaeogeography at 20 Ma
- Recent (quaternary) lakes
- Alluvial fans
- Inferred coastline
- Beach environment
- Estuarian environment
- Foreshore to Nearshore
- Subtidal shoals
- Mean transport directions

Vosges

Black Forest

Rhein-Bresse Graben

Basel

Recent Molasse basin border

Zurich

Besançon

Jura mountains

Hörnli-fan

Napf-fan

Aar-massif

Go

Fribourg-fans

Lepontine dome

Simplon fault

Geneva

Ancient Alpine thrust

Helveticum

AR

MB

Penninicum

geophysical Ivrea body

Southern Alps

50 km

N

[Figure]

[Figure]

**Figure 6:** Palaeogeographical reconstructions of the Molasse basin at different stages: **a)** USM (c. 22 Ma), **b)** OMM-Ia (c. 20 Ma), **c)** OMM-Ib (c. 19-18 Ma) and **d)** OMM-II to OSM (c. 18– df – 14 Ma) modified after Kuhlemann and Kempf (2002) and based on own observations. Note that all maps show present-day lithotectonic units within the Alps and the Jura mountains for orientation purposes (dashed lines and grey-coloured lines). We acknowledge that the positions of these and the surface patterns (such as lakes) were different during deposition of the Molasse deposits. The location of the palaeo-thrust fronts (thick linelines) are adapted from Kuhlemann and Kempf (2002). Black dots mark study sites for orientation purposes. Please refer to Fig. 1 for the complete legend.

[Figure]

Situation of the Alps at c. 20 - 18 Ma

Mod. after Schlunegger and Kissling (2015) and Kissling and Schlunegger (2018)

[Figure]

**Figure 7:** Simplified geological-geophysical model of the Alpine orogen for the time between 20 – 18 Ma, showing the most important geodynamic forces that might have shaped the Molasse basin and induced the transgression. Modified after Schlunegger and Kissling (2015) and Kissling and Schlunegger (2018).

[Figure]

[Figure]

**Figure 8:** Molasse stages (USM, OMM and OSM) with mean palaeo-transport directions (black arrows), hiatuses plotted against stable oxygen isotope stages (Miller et al., 1996) and sediment flux (Kuhlemann, 2000; Willet, 2010). The red triangle demarcates the onset of delamination and rapid exhumation of the Aar-massif (Herwegh et al., 2017). Grey arrows demarcate  decreases in sediment flux and falls in the eustatic sea level possibly contributing to the related hiatuses. MFS = Maximum-flooding surface.